# Type-I nNOS neurons orchestrate cortical neural activity and vasomotion

Kevin Turner[1,2,3,4], Dakota Brockway[4,5], Md Shakhawat Hossain[2,3,4], Keith Griffith[4,5], Denver Greenawalt[4,6], Qingguang Zhang[1,2,4,7], Kyle Gheres[1,2,4], Nicole Crowley[2,3,4,5], Patrick J Drew[1,2,3,4,5,8]*

[1]Department of Engineering Science and Mechanics, The Pennsylvania State University, University Park, United States; [2]Center for Neural Engineering, The Pennsylvania State University, University Park, United States; [3]Department of Biomedical Engineering, The Pennsylvania State University, University Park, United States; [4]Penn State Neuroscience Institute, The Pennsylvania State University, University Park, United States; [5]Department of Biology, The Pennsylvania State University, University Park, United States; [6]Graduate Program in Molecular Cellular and Integrative Biosciences, The Pennsylvania State University, University Park, United States; [7]Department of Physiology, Michigan State University, East Lansing, United States; [8]Department of Neurosurgery, The Pennsylvania State University, University Park, United States

*For correspondence: PJD17@PSU.EDU

## eLife Assessment

This **important** study provides **solid** evidence for new insights into the role of Type-1 nNOS interneurons in driving neuronal network activity and controlling vascular network dynamics in awake, head-fixed mice. The authors use an original strategy based on the ablation of Type-1 nNOS interneurons with local injection of saporin conjugated to a substance P analogue into the somatosensory cortex. They show that ablation of type I nNOS neurons has surprisingly little effect on neurovascular coupling, although it alters neural activity and vascular dynamics.

**Abstract** It is unknown how the brain orchestrates coordination of global neural and vascular dynamics. We sought to uncover the role of a sparse but unusual population of genetically distinct interneurons known as type-I nNOS neurons, using a novel pharmacological strategy to unilaterally ablate these neurons from the somatosensory cortex of mice. Region-specific ablation produced changes in both neural activity and vascular dynamics, decreased power in the delta-band of the local field potential, reduced sustained vascular responses to prolonged sensory stimulation, and abolished the post-stimulus undershoot in cerebral blood volume. Coherence between the left and right somatosensory cortex gamma-band power envelope and blood volume at ultra-low frequencies was decreased, suggesting type-1 nNOS neurons integrate long-range coordination of brain signals. Lastly, we observed decreases in the amplitude of resting-state blood volume oscillations and decreased vasomotion following the ablation of type-I nNOS neurons. This demonstrates that a small population of nNOS-positive neurons is indispensable for regulating both neural and vascular dynamics in the whole brain, raising the possibility that loss of these neurons could contribute to the development of neurodegenerative diseases and sleep disturbances.

## Introduction

Coordinated neural and hemodynamic activity across the brain is vital for arousal and cognition. The relationship between changes in neural activity and (typically measured) changes in blood volume, flow, and oxygenation, which are frequently studied with hemodynamic imaging and form the foundation of modern neuroscience, is mediated by local vasodilation and is known as neurovascular coupling (*Drew, 2019*; *Schaeffer and Iadecola, 2021*). Neurovascular coupling plays a critical role in supporting neuronal function, as tightly coordinated hemodynamic activity is essential for meeting energy metabolism and maintaining brain health (*Iadecola et al., 2023*; *Schaeffer and Iadecola, 2021*). The vasodilator nitric oxide (NO) (*Hosford and Gourine, 2019*), produced by neuronal nitric oxide synthase (nNOS) in neurons, provides a key mechanism of arteriole dilation. Neurons in somatosensory cortex that express nNOS are activated by sensory stimulation, volitional whisking, and locomotion (*Ahn et al., 2023*; *Echagarruga et al., 2020*), all of which produce robust hemodynamic responses (*Huo et al., 2014*; *Winder et al., 2017*). Chemogenetic elevation or suppression of nNOS neuron activity alters both baseline diameter and behaviorally evoked dilation of arterioles (*Echagarruga et al., 2020*). Similar vascular changes are produced by local infusion of an NOS inhibitor (*Echagarruga et al., 2020*), implicating NO signaling directly in vascular regulation. Optogenetic activation of nNOS-positive neurons produces varying effects, with some reports demonstrating vasodilation and blood flow increases with minimal changes in neural activity (*Krawchuk et al., 2020*; *Lee et al., 2020*; *Ruff et al., 2024*), while others show large, low-frequency EEG changes during sleep and quiet wakeful states (*Gerashchenko et al., 2018*). While NO is known to modulate the excitability of neurons (*Kara and Friedlander, 1999*; *Smith and Otis, 2003*), artificial activation of nNOS neurons seems to drive vascular responses that are disproportionally larger than any corresponding neural changes, suggesting that this small population of neurons has an impact on vascular signals that is substantially greater than that of overall neural activity.

Two distinct GABAergic interneuron types express nNOS, denoted as type-I and type-II (*Kawaguchi and Kubota, 1997*; *Kubota et al., 2011*; *Perrenoud et al., 2012*). nNOS-positive neurons make up approximately 20% of GABAergic interneurons in the cortex (*Hendry et al., 1987*; *Sahara et al., 2012*), with type-I constituting only 0.5–2% depending on brain region (*Chong et al., 2019*; *Tricoire and Vitalis, 2012*; *Yan and Garey, 1997*). Type-I nNOS neurons are sparse, somatostatin-positive, most dense in the deep layers of cortex, and express nNOS at higher levels than type-II. They extend both local and long-range projections throughout the cortex, receive excitatory input from nearby pyramidal neurons, and drive increases in cerebral blood flow when stimulated (*Ruff et al., 2024*). Type-II nNOS neurons are, in contrast, much more abundant, smaller in size and morphology, and express nNOS less strongly than type-I. Type-II are a more heterogeneous group that express a variety of interneuron subtype markers and are more evenly distributed throughout the different layers of cortex (*Perrenoud et al., 2012*; *Yan et al., 1996*). Type-I nNOS neurons have been observed in rodents, carnivores, and primates (*Higo et al., 2007*; *Tomioka et al., 2005*; *Tomioka and Rockland, 2007*) and are active during sleep (*Dittrich et al., 2015*; *Gerashchenko et al., 2008*; *Kilduff et al., 2011*; *Morairty et al., 2013*). In the cortex, type-I nNOS neuron density is anticorrelated with vascular density (*Wu et al., 2022*). Type-I nNOS neurons integrate local activity from feedforward excitatory pathways from the cortex and thalamus, likely contributing to the correlated changes in blood flow that are observed between functionally connected regions across hemispheres (*Ruff et al., 2024*). Molecular, electrophysiological, and immunohistochemistry studies have demonstrated that type-I nNOS neurons are the only cells in the cortex that express the tachykinin receptor 1 (also known as neurokinin 1 receptor, TACR1/NK1R), the primary receptor for the endogenous neuropeptide substance P (*Dittrich et al., 2012*; *Endo et al., 2016*; *He et al., 2016*; *Huang et al., 2016*; *Kubota et al., 2011*; *Matsumura et al., 2021*; *Paul et al., 2017*; *Penny et al., 1986*; *Ruff et al., 2024*; *Vanlandewijck et al., 2018*; *Vruwink et al., 2001*). Local infusion of substance P into the cortex causes a sustained increase in basal arterial diameter that is dependent upon local neural activity (*Echagarruga et al., 2020*). Type-I nNOS neurons likely receive substance P from the ~40% of parvalbumin (PV)-positive cortical interneurons (*Bugeon et al., 2022*; *Pfeffer et al., 2013*; *Vruwink et al., 2001*) that are thought to drive network synchrony (*Cardin et al., 2009*; *Sohal et al., 2009*). Optogenetic stimulation of PV neurons for several seconds drives a biphasic hemodynamic response, comprised of an early constriction (driven by suppression of overall population activity through GABA release), and a delayed, prolonged dilation lasting tens of seconds that is mediated indirectly by substance P

(*Vo et al., 2023*). The delayed vasodilation was not directly related to pyramidal neuron activity, was blocked by TACR1 antagonists, and was occluded by substance P, suggesting that activation of PV neurons drives downstream activation of type-I nNOS neurons. These experiments point to type-I nNOS neurons as having a large effect on vascular dynamics despite their smaller contribution in driving overall neural activity. However, optogenetic stimulation only activates one component of a circuit in isolation, in an otherwise healthy and intact microcircuit, potentially giving an incomplete picture of the neuron's role. Type-I nNOS neurons are uniquely vulnerable to stress (*Han et al., 2019*), and their loss is likely in part causal to neurodegeneration (*Iadecola et al., 2023*; *Schaeffer and Iadecola, 2021*), making it vital to understand brain-wide changes following a more physiologically relevant model of perturbation.

Here, we sought to reveal the role of type-I nNOS neurons in controlling neural and vascular dynamics in the cortex by selectively ablating them with saporin conjugated to a substance P analog (SP-saporin). Using this targeted ablation approach, we found that ablation of type-I nNOS neurons caused decreases in the hemodynamic response to sustained vibrissae stimulation, eliminated the post-stimulus undershoot, decreased local field potential (LFP) power in the delta-band (1–4 Hz), reduced bilateral correlations in gamma-band power and blood volume across arousal states, and reduced the amplitude of resting-state blood volume oscillations. Together, these experiments demonstrate that a small subset of type-I nNOS neurons regulate key neural and vascular dynamics that coordinate changes in vasomotion.

## Results

We investigated the effects of localized ablation of type-I nNOS neurons in somatosensory cortex on neurovascular coupling and functional connectivity in unanesthetized, head-fixed mice. C57BL6J mice (119 total, both male and female) were injected with either saporin conjugated to a substance P analog with high affinity for the substance P receptor (SP-SAP) or a scrambled peptide as a control (Blank-SAP) into a localized region of one hemisphere's somatosensory cortex. We used widefield optical imaging, electrophysiology, and two-photon microscopy to evaluate neural and hemodynamic changes following targeted ablation of type-I nNOS neurons in the vibrissae (whisker) representation of somatosensory cortex while carefully monitoring arousal state (see Materials and methods). All imaging was performed during the animals' light cycle. Statistics included generalized linear mixed-effects model (GLME), general linear models (GLM), or unpaired *t*-tests with corrections for multiple comparisons when necessary.

### Saporin-conjugated peptides produce selective targeted ablation of type-I nNOS neurons

While optogenetic and chemogenetic models give us insights into the potential function of neuron subtypes, it is difficult to completely silence neurons in vivo, and the patterns of activity they induce are not physiological and can have paradoxical effects on neural activity (*Andrei et al., 2021*; *Li et al., 2019*). Saporin provides a pharmacological route to selectively kill only the cells that internalize it, and saporin conjugated to a peptide effectively targets neurons that express a receptor for the peptide (*Abbott et al., 2012*; *McKay and Feldman, 2008*), allowing targeting of specific cell types orthogonal to any genetic targeting techniques.

We first sought to validate the efficacy and specificity of saporin-based targeting of cortical TACR1-expressing neurons. Type-I nNOS neurons were ablated by injection of the ribosome inactivating protein saporin (SAP), conjugated either to substance P (SP-SAP) or a scrambled peptide as a control (Blank-SAP) (*Figure 1a*). The SP-bound SAP toxin binds to TACR1-expressing neurons (*Martin and Sloviter, 2001*; *Wang et al., 2002*; *Wiley and Lappi, 1999*), which in the cortex are exclusively expressed by type-I nNOS neurons (*Figure 1b*). We used both immunofluorescence (*Figure 1c*) and NADPH diaphorase staining (a histochemical marker for NOS, *Figure 1—figure supplement 1*), to visualize nNOS neuron ablation (*Bredt et al., 1991*; *Hope et al., 1991*; *Hope and Vincent, 1989*). Immunofluorescent labeling in mice injected with Blank-SAP showed labeling of nNOS-positive neurons near the injection site. In contrast, mice injected with SP-SAP showed a clear loss in nNOS labeling, with a typical spread of 1–2 mm from the injection site, though nNOS-positive neurons both subcortically and in the entirety of the contralateral hemisphere remained intact. We quantified the

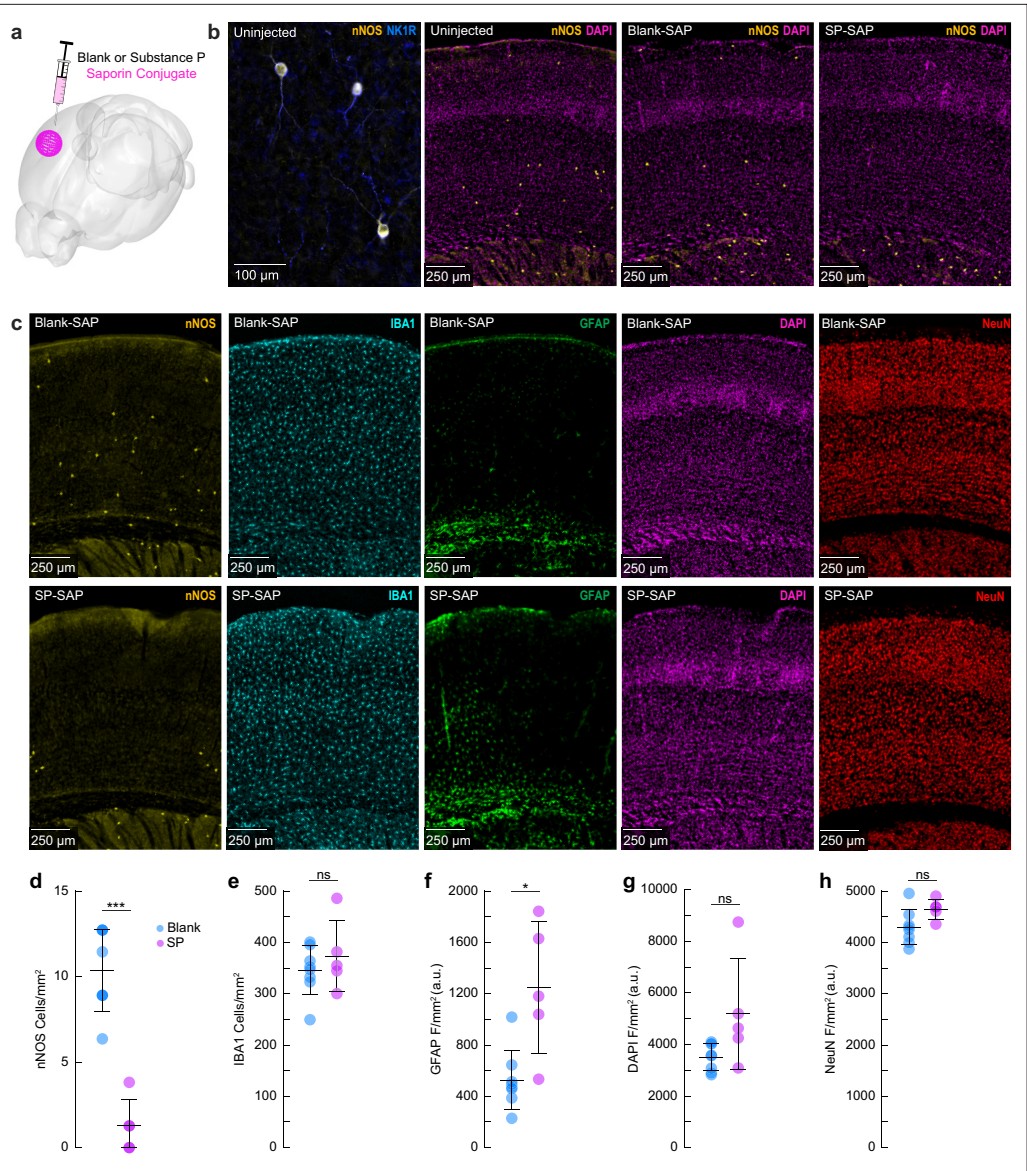

**Figure 1.** Saporin-conjugated peptides produce selective targeted ablation of type-I nNOS neurons. (**a**) Schematic showing intracortical administration of the ribosome inactivating protein saporin conjugated to either SP or a vehicle control. (**b**) Representative immunofluorescence of anti-nNOS (yellow) and anti-NKR1 (blue) showing colocalization of the NK1 receptor on cortical nNOS-positive neurons. Representative immunofluorescence of anti-nNOS (yellow) and DAPI (magenta) from animals with no injection (Uninjected, left), Blank-SAP (middle), or SP-SAP (right). (**c**) Representative immunofluorescence of Blank-SAP (top) and SP-SAP (bottom) sections co-stained with nNOS (left), IBA1 (middle left), GFAP (middle), DAPI (middle right), or NeuN (right). NeuN image was taken from the section immediately adjacent to the first four. (**d**) Quantification of nNOS counts, (**e**) IBA1 counts, (**f**) GFAP fluorescence, (**g**) DAPI fluorescence, and (**h**) NeuN fluorescence. Error bars (**d–h**) denote SD. Bonferroni correction (5) *$\alpha < 0.01$, ***$\alpha < 0.002$, GLME.

The online version of this article includes the following figure supplement(s) for figure 1:

**Figure supplement 1.** Histological quantification of cortical SP-SAP injections.

**Figure supplement 2.** Behavior was unaffected by local ablation of type-I nNOS neurons.

**Figure supplement 3.** Sleep classification accuracy was unchanged following type-I nNOS ablation (**a**).

efficacy of SP-SAP in removing type-I nNOS neurons by quantifying the number of nNOS-positive neurons per square mm of cortical tissue (*Figure 1d*), of which SP-SAP injected mice (*N* = 5, 3M/2F) had a significant reduction of nNOS-labeled neurons (1.3 ± 1.6 neurons/mm$^2$) compared to Blank-SAP mice (*N* = 8, 4M/4F) with 10.3 ± 2.4 neurons/mm$^2$ (GLME p = 5.74 × 10$^{-6}$). It is unlikely that the disappearance of type-I nNOS neurons is because they stopped expressing nNOS, as internalized saporin is cytotoxic. Exposure to SP-conjugated saporin causes rapid internalization of the SP receptor–ligand complex (*Mantyh et al., 1995*), and internalized saporin causes cell death via apoptosis (*Bergamaschi et al., 1996*). In the brain, the resulting cellular debris from saporin administration is then cleared by microglia phagocytosis (*Seeger et al., 1997*).

We checked for non-specific effects driven by SP-SAP injections 1-month post-administration by staining for ionized calcium binding adaptor molecule 1 (IBA1), glial fibrillary acidic protein (GFAP), DAPI, and NeuN (neuronal nuclei) as well as nNOS (as a positive control for ablation). There was no difference in the number of microglia (*Figure 1e*) between groups (Blank-SAP: 345.8 ± 47.7 microglia/mm$^2$; SP-SAP: 373.8 ± 69.4 microglia/mm$^2$). There was an increase in GFAP labeling following SP-SAP injection (1246.3 ± 514.8 AU/mm$^2$; Blank-SAP: 527.2 ± 230.9 AU/mm$^2$, GLME p = 0.0029) (*Figure 1f*). There was no significant difference in DAPI fluorescence (Blank-SAP: 3511.2 ± 524.42 AU/mm$^2$; SP-SAP: 5180.7 ± 2135.0 AU/mm$^2$; GLME p = 0.038, *Figure 1g*) or NeuN fluorescence (Blank-SAP: 4296.8 ± 339.4 AU/mm$^2$; SP-SAP: 4652.1 ± 196.8 AU/mm$^2$; GLME p = 0.043, *Figure 1h*) between groups after Bonferroni correction for (5) multiple comparisons (*Figure 1d–h*). Together, these findings indicate that the SP-SAP toxin was highly selective in ablating type-I nNOS neurons with minimal non-specific effects. All mice that underwent imaging were histologically verified for successful type-I nNOS neuron ablation using NADPH diaphorase staining. We quantified a subset of these mice and saw a similar specific removal of type-I nNOS neurons (*Figure 1—figure supplement 1*). To determine if ablation of type-I nNOS neurons from the somatosensory cortex had any impact on behavior or arousal state, we assayed exploratory behavior, sleep, and pupil dynamics (*Figure 1—figure supplements 2 and 3*). To assay exploratory behavior, mice were placed in a novel open field environment and allowed to explore for 5 min while quantifying distance traveled and time spent in the center, two metrics for evaluating stress and anxiety in rodents (*Roth and Katz, 1979*; *Seibenhener and Wooten, 2015*). We noted no differences in any metric evaluating open field behavior (*N* = 12–23 mice/group). We also assessed multiple measurements of sleep quality and quantity. There was no difference in the percentage of time each mouse spent in rapid eye movement (REM) or non-REM (NREM) sleep, or the percentage of time they spent volitionally whisking while awake (*N* = 9 mice/group). We also noted no significant differences in eye-related arousal state metrics including interblink interval, pupil size, and pupillary response to vibrissae stimulation (*Turner et al., 2023*). Together, this suggests that ablation of type-I nNOS neurons had no gross effects on sleep or general ambulatory behavior.

## Impact of type-I nNOS neuron removal on neural and hemodynamic signals across arousal states

To determine the impact localized ablation of type-I nNOS neurons had on neural and hemodynamic signals in the somatosensory cortex, we used widefield optical imaging (*Huo et al., 2014*; *Sirotin and Das, 2009*) to measure changes in total hemoglobin (Δ[HbT], an indicator of blood volume). We measured neural activity using either implanted electrodes (*Winder et al., 2017*) to measure changes in LFP (*Figure 2a, c, d*), or in a separate cohort of mice, pan-neuronal expression of the calcium indicator GCaMP7s (*Chan et al., 2017*; *Dana et al., 2019*), which provides complementary measures of bulk neural activity (*Figure 2b, e, f*). In mice expressing GCaMP7s, we measured Δ[HbT], as well as changes in cerebral oxygenation (Δ[HbO]) and deoxygenation (Δ[HbR]) using alternating illumination at 480/530/630 nm (*Ma et al., 2016a*; *Ma et al., 2016b*; *Zhang et al., 2019*). We corrected for hemodynamic contamination of GCaMP7s signals using the simultaneously acquired hemoglobin signals (*Kramer and Pearlstein, 1979*; *Ma et al., 2016a*; *Scott et al., 2018*; *Wright et al., 2017*). Measurements of neural and hemodynamic signals were taken bilaterally through polished and reinforced thinned-skull windows (*Drew et al., 2010*; *Shih et al., 2012*) in the vibrissae representation of somatosensory cortex. Each animal was habituated to head fixation over the course of several days following surgery. For the first 60 min of each recording session, the vibrissae were briefly stimulated with directed puffs of air to either the left or right pad, or by a puffer directed away from the mouse as an auditory control (*Drew et al., 2011*). Afterward, each mouse was given several hours to naturally

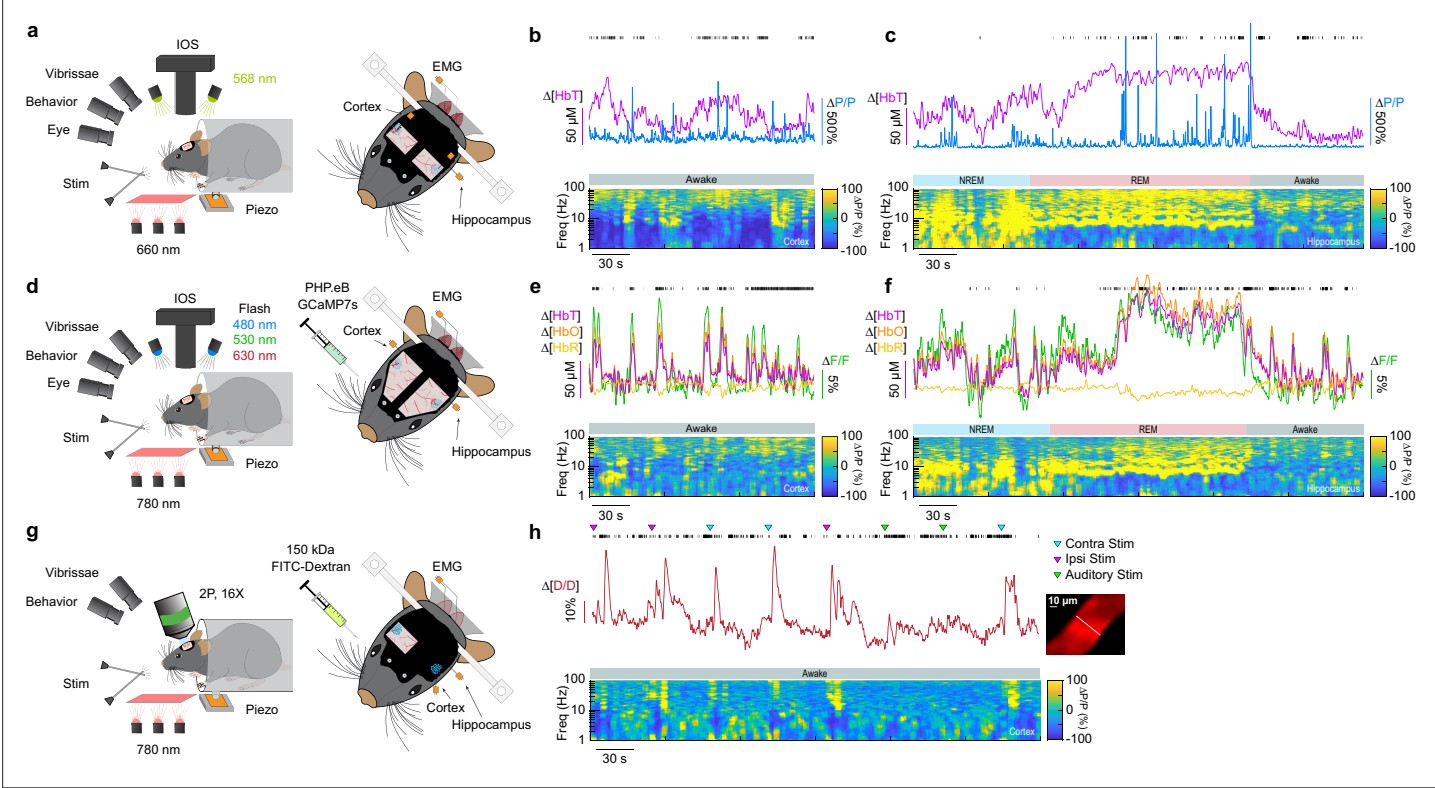

**Figure 2.** Impact of type-I nNOS neuron removal on neural and hemodynamic signals across arousal states. (**a**) Schematic of widefield optical imaging experimental setup. The brain is illuminated at an isosbestic wavelength of hemoglobin (568 nm). Changes in reflected light measuring changes in total blood volume are captured by a camera mounted above the head while several other cameras monitor animal behavior and arousal state including vibrissae and pupil tracking. Vibrissae stimulation is done by directed air puffs. Polished and reinforced thinned-skull windows were bilaterally implanted over the somatosensory cortex. Tungsten stereotrodes were implanted underneath each window to record changes in local field potential (LFP) within the area of interest (relative change in power, ΔP/P). An additional hippocampal stereotrode and a neck electromyography electrode were used to assist in sleep scoring. (**b, e, h**) During the awake state, power in low-frequency cortical LFP is low and power in the gamma-band (30–100 Hz) is elevated during activity such as volitional whisking. (**c, f**) Periods of non-rapid eye movement (REM) and rapid eye movement (REM) sleep are accompanied by large oscillations in cerebral blood volume, with large increases in power in delta-band (1–4 Hz) cortical LFP during NREM and large increases in theta-band (4–10 Hz) hippocampal LFP during REM. Black tick marks indicate vibrissae motion. (**d**) Schematic of widefield optical imaging experiments with GCaMP7s. Alternating illumination with 480, 530, and 630 nm light was used to measure changes in total hemoglobin, blood oxygenation, and GCaMP7s fluorescence. (**g**) Schematic of two-photon experiments.

sleep with no stimulation. To determine the arousal state of the mouse, we performed electromyography (EMG) of the nuchal muscles of the neck, tracked vibrissae movement and pupil diameter using video, and body movements with a force sensor. Arousal state was scored in 5-s intervals as either Awake, NREM, or REM as previously described (*Turner et al., 2020*) from behavioral and physiological data using a bootstrapped random forest classification algorithm. We saw no differences in the accuracy or validity of our sleep-scoring models across different experimental conditions (Blank-SAP; SP-SAP; Uninjected) (*Figure 1—figure supplement 3*).

## Ablation of type-I nNOS neurons reduces the stimulus-evoked response

We first determined the impact of removal of type-I nNOS neurons on evoked hemodynamic signals. The initial increase in blood volume in response to brief (0.1 s) stimulation of the contralateral vibrissae was not affected by ablation of type-I nNOS neurons (*Figure 3a*), but the post-stimulus undershoot was absent in the SP-SAP mice. When we evaluated the canonical post-stimulus hemodynamic undershoot from 2:4 s, the Blank-SAP group (*N* = 9, 5M/4F) had a mean of –2.2 ± 0.5 µM, compared to the SP-SAP group (*N* = 9, 5M/4F) mean of 1.0 ± 0.6 µM (GLME p = 0.0005). This result is consistent with the observation that type-I nNOS neurons express the vasoconstrictory neuropeptide Y (NPY) (*Karagiannis et al., 2009*) which is thought to underlie this post-stimulus undershoot (*Uhlirova et al., 2016*).

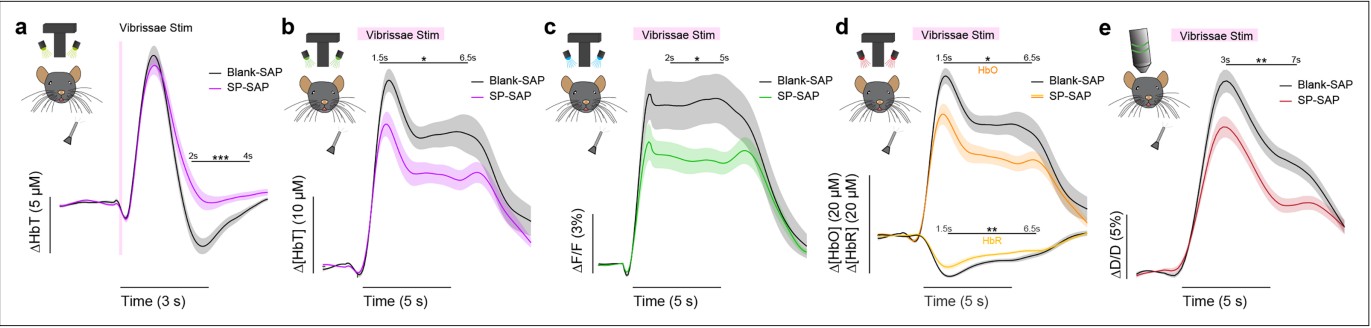

**Figure 3.** Ablation of type-I nNOS neurons reduces the stimulus-evoked response. (**a**) Change in total hemoglobin in response to brief (0.1 s) vibrissae stimulation. (**b**) Change in total hemoglobin in response to extended (5 s) vibrissae stimulation. (**c**) Change in GCaMP fluorescence in response to extended vibrissae stimulation. (**d**) Change in oxy- and deoxy-hemoglobin in response to extended vibrissae stimulation. (**e**) Change in arteriole diameter in response to extended vibrissae stimulation. Error bars represent population averages ± SEM. All statistics were evaluated between the indicated intervals. *p < 0.05, **p < 0.01, ***p < 0.001, GLME.

It is also possible that loss of NO signaling from type-I nNOS neurons could also contribute through interactions with blood volume to generate oscillations and post-dilation undershoots (*Haselden et al., 2020*). Using three-wavelength spectroscopy in mice expressing GCaMP7s, we saw consistent increases in blood volume, neural activity, and cerebral oxygenation during prolonged (5 s) vibrissae stimulation. Δ[HbT] evaluated 1.5:6.5 s following stimulus-onset decreased from 18.7 ± 1.7 μM in Blank-SAP (*N* = 7, 3M/4F) to 13.1 ± 1.4 μM in SP-SAP (*N* = 8, 4M/4F, GLME p = 0.017) (*Figure 3b*). Calcium signals (*Figure 3c*) evaluated 2:5 s following stimulus onset decreased from 8.2 ± 1.3% in Blank-SAP to 5.3 ± 0.7% in SP-SAP (GLME p = 0.047). Δ[HbO] (*Figure 3d*) evaluated 1.5:6.5 s following stimulus onset decreased from 24.6 ± 1.9 μM in Blank-SAP to 17.4 ± 1.7 μM in SP-SAP (GLME p = 0.010) while Δ[HbR] evaluated from 1.5:6.5 s increased from –6.0 ± 0.4 in Blank-SAP to –4.4 ± 0.3 in SP-SAP (GLME p = 0.005). Evaluation windows were determined based on the duration of stimuli and the delayed onset of the hemodynamic response.

We further investigated single arterial dynamics during vibrissae stimulation with two-photon microscopy. Following ablation of type-I nNOS neurons, after vibrissae stimulation there was a decrease in arteriole diameter (*Figure 3e*) evaluated 3:7 s following stimulus onset from 11.6 ± 0.8% s in Blank-SAP (*N* = 9, 5M/4F, *n* = 81 arterioles) down to 7.8 ± 0.6% in SP-SAP (*N* = 7, 2M/5F, *n* = 70 arterioles, GLME p = 0.008). The arterial changes closely mirror the Δ[HbT] changes, consistent with the substantial contribution of arterial dynamics to the blood volume signal (*Huo et al., 2015*). We also evaluated changes in blood volume during voluntary locomotion, where we saw a similar trend in

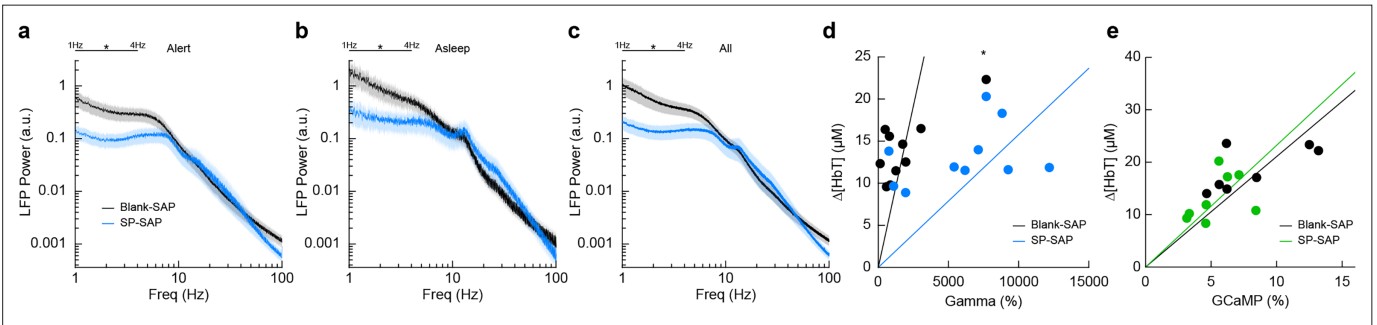

**Figure 4.** Type-I nNOS ablation alters low-frequency neural activity and gamma-band neurovascular coupling. (**a**) Local field potential within the vibrissae representation of somatosensory cortex during periods of Alert. (**b**) Asleep. (**c**) All data. (**d**) Change in gamma-band power versus Δ[HbT] following brief (0.1 s) vibrissae stimulation. (**e**) Change in GCaMP7s fluorescence versus Δ[HbT] following prolonged (5 s) vibrissae stimulation. Error bars represent population averages ± SEM. All statistics were evaluated between the indicated intervals. **p < 0.01, GLME (**a–c**), GLM (**d, e**).

The online version of this article includes the following figure supplement(s) for figure 4:

**Figure supplement 1.** Ablation of Type-I nNOS neurons does not alter local field potential (LFP) or vasomotion in the contralateral hemisphere.

**Figure supplement 2.** Ablation of type-I nNOS neurons does not alter hemodynamic or neural power spectra.

the decrease in locomotion-evoked Δ[HbT] evaluated 1.5:2.5 s following locomotion onset from 17.5 ± 2.0 μM in Blank-SAP (N = 7, 5M/2F) to 12.9 ± 2.0 μM in SP-SAP (N = 7, 2M/5F), but it was not statistically significant (GLME p = 0.10). These results show the removal of a very small number of neurons can drive substantial reductions in both hemodynamic and neural response to sensory stimulation.

## Type-I nNOS ablation reduces low-frequency neural activity and gamma-band neurovascular coupling

To determine the effect that type-I nNOS neuron removal had on neural activity, we assessed the power spectrum of the LFP in different arousal states, which we classified into alert, asleep, and all data (*Turner et al., 2020*). Data was classified into the alert state when 15-min blocks were predominantly (>80%) in the awake state. Alert periods contained fidgeting movements and bouts of whisking interspersed with awake quiescence. Periods classified as asleep were predominantly composed of sleeping states (NREM/REM >80% of all classifications in a 15-min block) with only brief periods of wakefulness, typically occurring during transitions between sleep states. The 'all' state includes everything irrespective of arousal state classification. The first ~60 min of each recording session which included vibrissae stimulation was excluded. There was a pronounced reduction in the delta-band (1–4 Hz) power of the LFP in all three arousal state categories. The power in the delta-band of the LFP in Blank-SAP mice (N = 9, 4M/5F) in the alert state (*Figure 4a*) was $3.4 \times 10^{-10} \pm 1.0 \times 10^{-10}$ a.u. compared to $1.0 \times 10^{-10} \pm 3.0 \times 10^{-11}$ a.u. in the SP-SAP mice (N = 9, 5M/4F, GLME p = 0.011). The power in the delta-band of the LFP in Blank-SAP mice (N = 7, 3M/4F) in the asleep state (*Figure 4b*) was (NREM + REM) was $8.5 \times 10^{-10} \pm 3.0 \times 10^{-10}$ a.u. compared to $2.4 \times 10^{-10} \pm 9.1 \times 10^{-11}$ a.u. in the SP-SAP mice (N = 7, 4M/3F, GLME p = 0.022). The power in the delta-band of the LFP in Blank-SAP mice (N = 9, 4M/5F) averaged across all arousal states (*Figure 4c*) was $5.2 \times 10^{-10} \pm 1.7 \times 10^{-10}$ a.u. compared to $1.5 \times 10^{-10} \pm 4.3 \times 10^{-11}$ a.u. in the SP-SAP mice (N = 9, 5M/4F, GLME p = 0.016). These results parallel those seen in studies where knockout of nNOS produces lower delta-band power (*Morairty et al., 2013*) and optogenetic stimulation of nNOS neurons produces low-frequency oscillations (*Zielinski et al., 2019*), suggesting that type-I nNOS neurons promote these slow oscillations via NO release. In contrast to the observed reductions in LFP in the ablated hemisphere, we noted no gross changes in the power spectra of neural LFP in the unablated hemisphere (*Figure 4—figure supplement 1*) or power of the cerebral blood volume fluctuations in either hemisphere (*Figure 4—figure supplement 2*).

We next wanted to see whether the amount of neural activity and corresponding Δ[HbT] in response to vibrissae stimulation was altered following type-I nNOS neuron ablation. As a measure of neurovascular coupling, we used the slope of the line fitting the change in gamma-band power vs. Δ[HbT] following brief (0.1 s) vibrissae stimulation (*Figure 4d*). Decreases in the slope indicate a smaller vascular response for a given amount of neural activity, indicating a decrease in neurovascular coupling. We found the slope was significantly increased in SP-SAP mice compared to Blank-SAP mice (Blank-SAP: 0.02 ± 0.03; SP: 0.005 ± 0.006, GLM p = 0.0001). We also evaluated the change in GCaMP7s fluorescence versus Δ[HbT] during prolonged (5 s) vibrissae stimulation from 2:5 s (*Figure 4e*) and saw no significant change in the slope (Blank-SAP: 251.3 ± 75.3; SP: 256.5 ± 73.2, GLM p = 0.34). Given that the sensory-evoked LFP is driven more by synaptic input, while GCaMP7s signals are driven by local neural activity, these results indicate that the loss of type-I nNOS neurons reduces overall excitability, but that the input drive from other areas might be increased, potentially due to homeostatic mechanisms at the input synapses (*Turrigiano, 2008*).

## Neurovascular coupling was weakly affected by type-I nNOS removal

We next looked at the cross-correlation between blood volume and neural signals, which provides a measure of spontaneous neurovascular coupling. For electrophysiological measures, we measured the cross-correlation between gamma-band power and blood volume (*Figure 5a*). We evaluated the peak cross-correlation during the resting state, during long periods while alert, or long periods while asleep. The cross-correlation between neural activity and hemodynamic signals is substantially higher during periods of behavior (whisking, fidgeting) than during the resting state because self-generated motion and whisking drive increases in neural activity and vasodilation (*Claron et al., 2023*; *Drew et al., 2019*; *Stringer et al., 2019*; *Tu et al., 2024*; *Winder et al., 2017*). The correlation between neural activity and blood volume changes is higher during sleep than in the awake state (*Turner et al., 2020*).

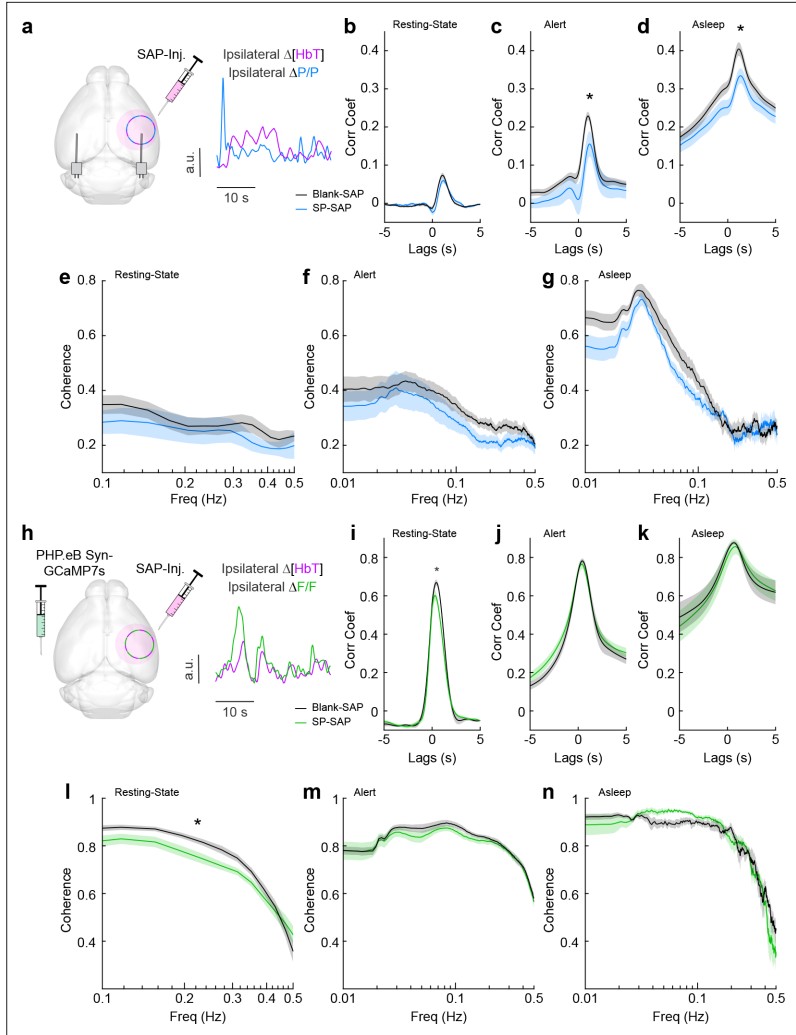

**Figure 5.** Neurovascular coupling was only weakly affected by type-I nNOS removal. (**a**) Schematic demonstrating intracortical injection of either Blank-SAP or SP-SAP and the analysis of gamma-band power and hemodynamic signals from within the vibrissae representation of somatosensory cortex, $N = 9$ mice per group. (**b**) Gamma-Δ[HbT] resting-state cross-correlation. (**c**) Gamma-Δ[HbT] alert cross-correlation. (**d**) Gamma-Δ[HbT] asleep cross-correlation. (**e**) Gamma-Δ[HbT] resting-state coherence. (**f**) Gamma-Δ[HbT] alert coherence. (**g**) Gamma-Δ[HbT] asleep coherence. (**h**) Schematic demonstrating intracortical injection of either Blank-SAP or SP-SAP and the analysis of GCaMP7s fluorescence and hemodynamic signals from within the vibrissae cortex, $n = 6–7$ mice per group. (**i**) GCaMP7s-Δ[HbT] resting-state cross-correlation. (**j**) GCaMP7s-Δ[HbT] alert cross-correlation. (**k**) GCaMP7s-Δ[HbT] asleep cross-correlation. (**l**) GCaMP7s-Δ[HbT] resting-state coherence. (**m**) GCaMP7s-Δ[HbT] alert cross-correlation. (**n**) GCaMP7s-Δ[HbT] asleep cross-correlation. Error bars represent population averages ± SEM. *$p < 0.05$, **$p < 0.01$, ***$p < 0.001$, GLME.

The online version of this article includes the following figure supplement(s) for figure 5:

**Figure supplement 1.** Ablation of type-I nNOS neurons does not alter the predictive power of the hemodynamic response function (HRF).

We saw no significant difference in the peak cross-correlation during the resting state (**Figure 5b**) between Blank-SAP mice ($N = 9$, 4M/5F, $0.08 ± 0.007$) and SP-SAP mice ($N = 9$, 5M/4F, $0.06 ± 0.008$, GLME $p = 0.22$). We did observe a significant difference in the peak cross-correlation during both periods of alert and asleep, from $0.23 ± 0.01$ (Blank-SAP) to $0.16 ± 0.03$ (SP-SAP, GLME $p = 0.036$) during alert periods (**Figure 5c**) and $0.40 ± 0.02$ (Blank-SAP) dropping to $0.33 ± 0.02$ (SP-SAP, GLME $p = 0.012$) while asleep (**Figure 5d**). We next wanted to evaluate whether this drop in neurovascular coupling was more prevalent at a particular modulation frequency. When analyzing the coherence

between the gamma-band power and hemodynamic signals during these different arousal states, we noted no dominant frequency and saw no significant changes in the lower frequencies associated with neurovascular coupling (≤0.5 Hz). During the resting state (*Figure 5e*), the average coherence between 0.1 and 0.5 Hz was 0.28 ± 0.008 in Blank-SAP mice and 0.23 ± 0.01 in SP-SAP mice (GLME p = 0.23). The average coherence between 0.01 and 0.5 Hz during alert periods (*Figure 5f*) was 0.29 ± 0.004 in Blank-SAP mice and 0.25 ± 0.005 in SP-SAP mice (GLME p = 0.06), and during asleep periods (*Figure 5g*) it was 0.34 ± 0.008 in Blank-SAP mice and 0.32 ± 0.007 in SP-SAP mice (GLME p = 0.46). Lastly, to evaluate any changes in the predictive power of the neural–hemodynamic relationship, we fit a hemodynamic response function (HRF) using the gamma-band power and hemodynamic response following periods of vibrissae stimulation. We observed that ablation of type-I nNOS neurons did not alter the predictive power of the HRF (*Figure 5—figure supplement 1*).

We next evaluated neurovascular coupling from optical measures of bulk activity using GCaMP7s (*Figure 5h*). The peak resting-state cross-correlation (*Figure 5i*) between GCaMP7s fluorescence and ongoing hemodynamics showed a significant decrease, 0.68 ± 0.02 with Blank-SAP versus 0.62 ± 0.02 with SP-SAP (GLME p = 0.038). However, there was no drop in peak cross-correlation during either alert or asleep as seen with the gamma-band power. During alert periods (*Figure 5j*), the peak GCaMP7s cross-correlation was 0.78 ± 0.12 with Blank-SAP and 0.77 ± 0.02 with SP-SAP (GLME p = 0.54), while during asleep periods (*Figure 5k*), it was 0.88 ± 0.008 with Blank-SAP and 0.86 ± 0.04 with SP-SAP (GLME p = 0.58). Like the peak in cross-correlation, the average resting-state coherence (*Figure 5l*) did show a significant drop across the lower frequencies, 0.75 ± 0.01 with Blank-SAP versus 0.70 ± 0.01 with SP-SAP (GLME p = 0.049). There was no drop in coherence during alert periods (*Figure 5m*) at 0.78 ± 0.005 with Blank-SAP and 0.77 ± 0.004 with SP-SAP (GLME p = 0.42) nor during asleep periods (*Figure 5n*) at 0.75 ± 0.009 with Blank-SAP and 0.74 ± 0.01 with SP-SAP (GLME p = 0.63). While there are small changes in neural–hemodynamic correlations, the differences in neurovascular coupling across arousal states were relatively small following localized removal of type-I nNOS neurons, meaning any effects of ablation on hemodynamic responses are primarily mediated by changes in neural activity.

## Type-I nNOS ablation reduces low-frequency interhemispheric coherence

Because type-I nNOS neurons send and receive many modulatory signals across a range of distances, they could help serve to coordinate neural and vascular dynamics across the brain. We tested this hypothesis by comparing how loss of type-I nNOS neurons changed the coherences of neural and hemodynamics signals between the left and right vibrissae cortex, which are generally highly correlated across all behaviors and frequencies (*Turner et al., 2020*). Neural activity is also bilaterally correlated, though less so than vascular dynamics. Average resting-state coherence between the left and right Δ[HbT] signals (*Figure 6a*) in the vibrissae cortex was 0.79 ± 0.005 for Blank-SAP and 0.73 ± 0.008 for SP-SAP mice (p = 0.014, GLME, *Figure 6b*). This reduction in coherence was seen over all frequencies (0–0.5 Hz) during both alert periods (*Figure 6c*), at 0.86 ± 0.002 for Blank-SAP and 0.81 ± 0.003 for SP-SAP (p = 7.4 × 10$^{-6}$, GLME), as well as asleep periods (*Figure 6d*) with 0.86 ± 0.003 for Blank-SAP and 0.79 ± 0.005 for SP-SAP (p = 0.0008, GLME). We then looked at the coherence of gamma-band power across hemispheres, where average coherence for bilateral gamma-band signals in the resting state was 0.19 ± 0.006 (Blank-SAP) and 0.19 ± 0.01 (SP-SAP, p = 0.88, GLME, *Figure 6e, f*). There was no significant difference during the alert periods (*Figure 6g*) at 0.22 ± 0.005 (Blank-SAP) and 0.20 ± 0.006 (SP-SAP, p = 0.43, GLME), but ablation of type-I nNOS neuros produced a reduction of coherence during asleep periods (*Figure 6h*) at 0.35 ± 0.009 (Blank-SAP) and 0.29 ± 0.009 (SP-SAP, p = 0.047, GLME).

Average resting-state coherence across bilateral Δ[HbT] signals (*Figure 6i*) taken from the vibrissae cortex was 0.83 ± 0.005 (Blank-SAP) and 0.78 ± 0.007 (SP-SAP, p = 0.021, GLME, *Figure 6j*). This reduction in coherence persisted across all frequencies during both alert periods (*Figure 6k*) at 0.92 ± 0.001 (Blank-SAP) and 0.90 ± 0.001 (SP-SAP, p = 0.012, GLME) as well as sleep periods (*Figure 6l*) at 0.90 ± 0.002 (Blank-SAP) and 0.83 ± 0.005 (SP-SAP, p = 0.004, GLME). Average resting-state coherence across bilateral GCaMP7s signals (*Figure 6m*) taken from the vibrissae cortex was 0.75 ± 0.009 (Blank-SAP) and 0.70 ± 0.009 (SP-SAP, GLME p = 0.12, *Figure 6n*). This reduction in coherence persisted across all frequencies during both alert periods (*Figure 6o*) at 0.88 ± 0.003 (Blank-SAP)

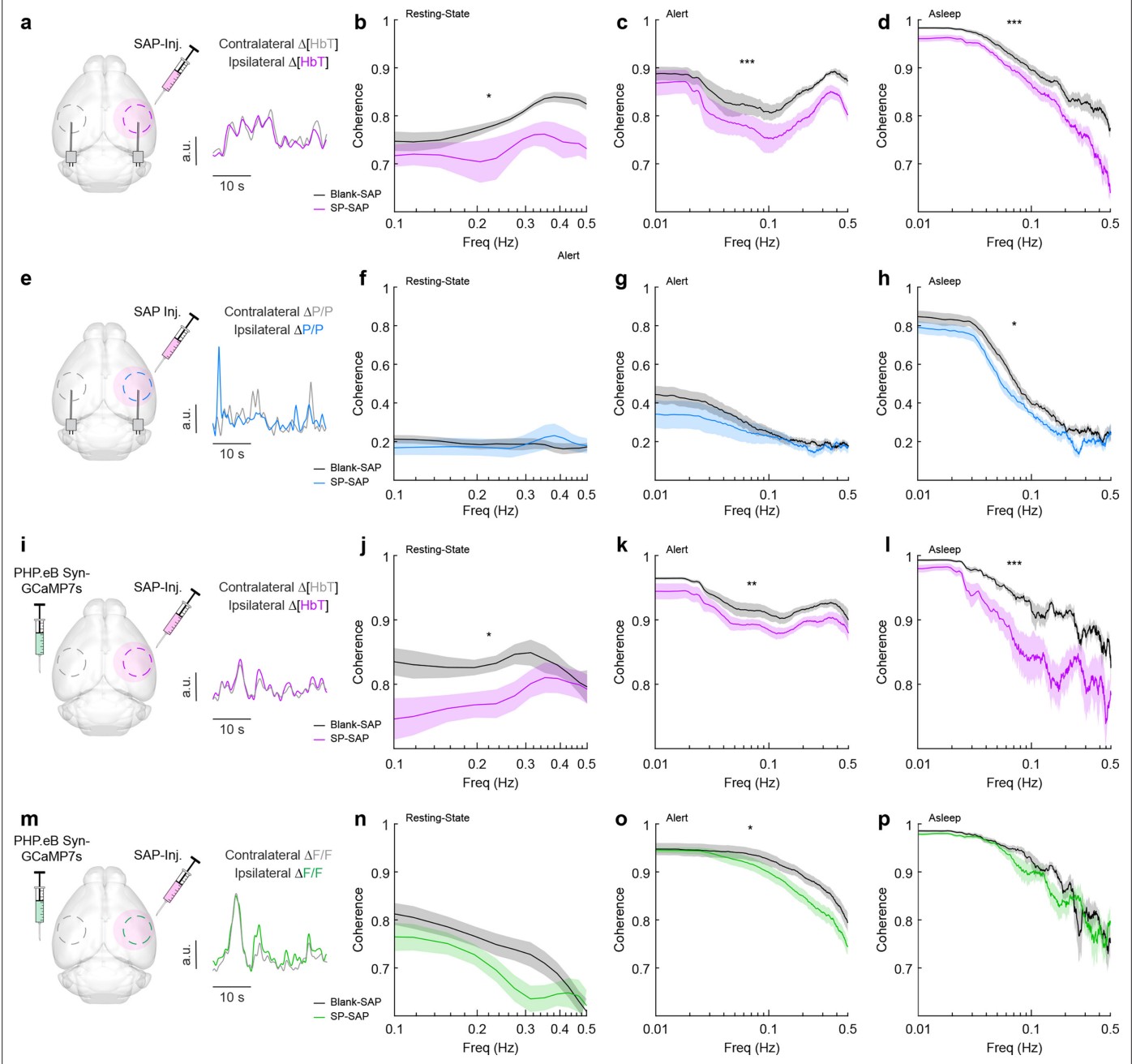

**Figure 6.** Type-I nNOS ablation reduces low-frequency interhemispheric coherence. (**a**) Δ[HbT] coherence between bilateral regions of interest (ROIs) in the left and right hemisphere's somatosensory cortex during (**b**) Rest, (**c**) Alert, and (**d**) Asleep. (**e**) Gamma-band power coherence between bilateral ROIs in the left and right hemisphere's somatosensory cortex during (**f**) Rest, (**g**) Alert, and (**h**) Asleep. (**i**) Δ[HbT] coherence between bilateral ROIs in the left and right hemisphere's somatosensory cortex during (**j**) Rest, (**k**) Alert, and (**l**) Asleep. (**m**) GCaMP7s coherence between bilateral ROIs in the left and right hemisphere's somatosensory cortex during (**n**) Rest, (**o**) Alert, and (**p**) Asleep. Error bars represent population averages ± SEM. *p < 0.05, **p < 0.01, ***p < 0.001, GLME.

The online version of this article includes the following figure supplement(s) for figure 6:

**Figure supplement 1.** Pearson's correlation coefficients between bilateral hemodynamic and neural signals.

and 0.84 ± 0.003 (SP-SAP, p = 0.029, GLME) but not in sleep periods (*Figure 6p*) at 0.85 ± 0.005 (Blank-SAP) and 0.84 ± 0.005 (SP-SAP, GLME p = 0.52). When the correlation of bilateral signals was evaluated with Pearson's correlation coefficients, the same general trend remained (*Figure 6—figure supplement 1*). These results show that removal of type-I nNOS neurons reduces both vascular and

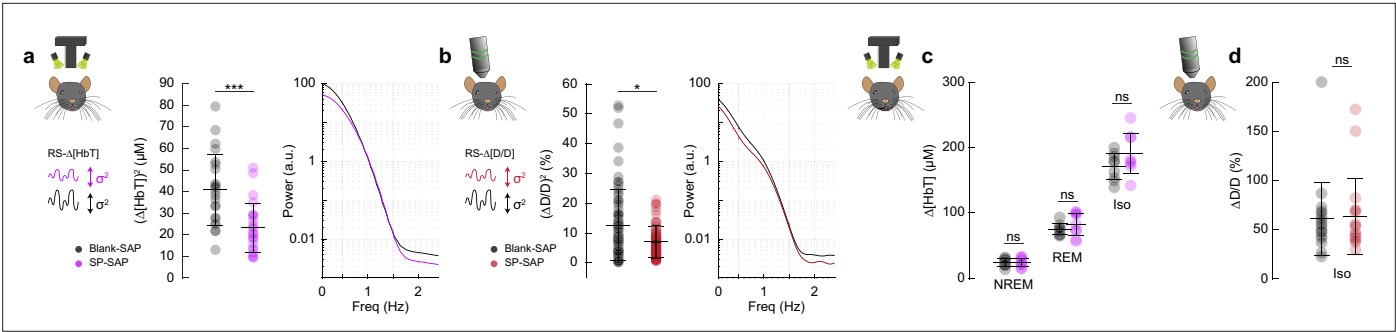

**Figure 7.** Type-I nNOS neurons control vasomotion power but not baseline diameter. (**a**) Variance (proportional to average power) in the resting-state hemodynamics signals measured with widefield optical imaging. (**b**) Variance in resting-state arteriole diameter measured with two photons. (**c**) Average Δ[HbT] during periods of non-rapid eye movement (NREM) sleep, periods of rapid eye movement (REM) sleep, and following administration of isoflurane (n = 9 mice per group). (**d**) Change in normalized arteriole diameter following administration of isoflurane. Error bars represent population averages ± SD. All statistics were evaluated between the indicated intervals. *p < 0.05, **p < 0.01, ***p < 0.001 GLME.

The online version of this article includes the following figure supplement(s) for figure 7:

**Figure supplement 1.** Removal of type-I nNOS neurons did not alter arousal state-related hemodynamic changes.

neural coordination across hemispheres. This loss of coherence does not imply that removal of type-I nNOS neurons has direct effects on the contralateral hemisphere, just that the dynamics in the ablated hemisphere have been altered so they no longer respond effectively to input from the ipsilateral hemisphere and/or common sources of modulatory drive that act through type 1 nNOS neurons (e.g. hypocretin, *Williams et al., 2019*, or cholinergic inputs, *Williams et al., 2018*).

## Type-I nNOS neurons control vasomotion power but not baseline diameter

We next wanted to establish how ablation of type-I nNOS neurons affected vasomotion, spontaneous oscillations in the absence of behavior, measured at the scale of blood volume and at the level of single arterioles, as well as the basal tone of blood vessels. The variance in Δ[HbT] during rest, a measure of vasomotion amplitude, was significantly reduced following type-I nNOS ablation (*Figure 7a*), dropping from 40.9 ± 3.4 $\mu M^2$ in the Blank-SAP group (N = 24, 12M/12F) to 23.3 ± 2.3 $\mu M^2$ in the SP-SAP group (N = 24, 11M/13F) (GLME p = 6.9 × $10^{-5}$) with no significant difference in the unablated hemisphere (Figure S7). Individual pial and penetrating arterioles showed the same reduction in vasomotion after type-I nNOS neuron ablation, with Blank-SAP (N = 9, 5M/4F, n = 70 arterioles) having a resting diameter variance of 12.6 ± 1.4$\%^2$ and SP-SAP (N = 7, 2M/5F, n = 65 arterioles) of 8.0 ± 0.8$\%^2$ (*Figure 7b*). There was no difference in resting-state (baseline) diameter between the groups, with Blank-SAP having a diameter of 24.4 ± 7.5 µm and SP-SAP having a diameter of 23.0 ± 9.4 µm (ttest, p = 0.61).

If type-I nNOS neurons control the basal diameter of vessels via secreted vasodilators, potentially in an arousal state dependent way, when we ablate these neurons, we would observe a difference in blood volume across arousal states and in the maximal dilation elicited with isoflurane (*Gao et al., 2015*). We saw no changes in the average change in Δ[HbT] across states (*Figure 7c*, *Figure 7—figure supplement 1*). The resting-state Δ[HbT] in Blank-SAP mice was 0.29 ± 0.7 µM compared to 0.29 ± 0.9 µM in SP-SAP (GLME p = 0.997). NREM Δ[HbT] in Blank-SAP mice was 23.9 ± 5.8 µM compared to 24.1 ± 6.5 µM in SP-SAP (GLME p = 0.96). REM Δ[HbT] in Blank-SAP mice was 74.5 ± 8.4 µM compared to 82.3 ± 17 µM in SP-SAP (GLME p = 0.26). The Δ[HbT] following isoflurane in Blank-SAP mice was 170.9 ± 19.7 µM compared to 190.5 ± 30.5 µM in SP-SAP (GLME p = 0.15). We repeated the isoflurane experiment on a set of pial and penetrating arterioles under two-photon microscopy. The change in normalized arteriole diameter following administration of isoflurane (n = 7–9 mice per group, 18–19 arterioles per group) was 61.2 ± 37.2% in Blank-SAP (baseline diameter 23.2 ± 6.6 µm) compared with 63.6 ± 38.4% in SP-SAP (baseline diameter 23.0 ± 10.2 µm, p = 0.85, ttest). These results indicate that type-I nNOS neurons play a role in driving spontaneous hemodynamic fluctuations, but not in setting the basal diameter during different states.

## Discussion

We selectively ablated type-I nNOS neurons from the somatosensory cortex, which had marked effects on neural activity and vascular dynamics, but minimal changes in neurovascular coupling. These results are surprising given previous work showing stimulation of these neurons in isolation causes vasodilation and minimal neural activity changes. Our approach of using SP-conjugated saporin allowed a nongenetic means of targeting a critical neuronal cell type, supporting further exploration of the role of type-I nNOS neurons in transgenic mouse models of disease without complicated breeding schemes, as well as in non-model organisms. Our results point to these neurons being an important orchestrator of neural and vascular dynamics, as loss of these neurons causes desynchronization between hemispheres

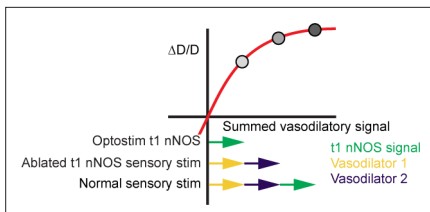

**Figure 8.** Schematic showing how non-linearity in the dilation response can explain coexistence of strong dilation by activation of a pathway and little change after weakening a pathway. If the diameter is a sublinear function of the sum of vasodilatory inputs, activation of all pathways will cause a dilation that is smaller than the sum of activation of each pathway individually. Loss of one pathway will not cause large changes, even though activation of that pathway in isolation can cause large dilations.

of both neural and vascular signals, as well as altered responses to sensory stimulation. While previous studies using specific activation of type-I nNOS neurons have emphasized the roles of these neurons in driving vasodilation with minimal changes in neural activity, our results point to a larger role of these neurons in organizing and patterning both spontaneous and sensory-evoked neural activity. This is likely due to a loss in NO signaling as well as a loss of many of the other neuropeptides expressed by type-I nNOS neurons, which all have known effects on other neurons. Loss of type-I nNOS neurons drove minimal changes in the vasodilation elicited by brief stimulation, but led to decreased vascular responses to sustained stimulation, suggesting that the early phase of neurovascular coupling is not mediated by these cells, consistent with the multiple known mechanisms for neurovascular coupling (*Attwell et al., 2010*; *Drew, 2019*; *Hosford and Gourine, 2019*) acting through both neurons and astrocytes with multiple timescales (*Le Gac et al., 2025*; *Renden et al., 2024*; *Schulz et al., 2012*; *Tran et al., 2018*).

While previous studies have found that activation of type-I nNOS neurons, either optogenetically or via administration of substance P, results in vasodilation (which would imply an important role for these neurons for neurovascular coupling), our work provides insight into the integrated function of type-I nNOS neurons in neural circuits and vascular dynamics. Importantly, methodological approaches differentiate our strategy from previous published work. A limitation of our study is the necessary delay between targeted ablation and in vivo imaging for the animals to recover from surgical procedures. With imaging beginning 4 weeks after ablation, there could be compensatory rewiring of local and/or network activity following type-I nNOS ablation, where other signaling pathways from the neurons to the vasculature become strengthened to compensate for the loss of vasodilatory signaling from the type-I nNOS neurons. While this likely happens to some degree, this interpretation is less likely to completely account for the effect given these potential compensatory changes do not prevent large changes in neural activity and resting vasomotion. Second, there may be some non-linear interactions between vasodilatory signals and/or the vasodilator mechanisms that are arousal-state-dependent. Intrinsic optical signal readout is primarily weighted toward superficial tissue given the absorption and scattering characteristics of the wavelengths used. While surface vessels are tightly coupled with neural activity, it is still a matter of debate whether surface or intracortical vessels are a more reliable indicator of ongoing activity (*Goense et al., 2012*; *Huber et al., 2015*; *Poplawsky and Kim, 2014*). The diameter of arteries tracks the smooth muscle membrane potential linearly, up until a saturation point above which any hyperpolarization does not induce further dilation (*Hill, 2012*; *Knot and Nelson, 1998*; *Wölfle et al., 2011*). Many vasoactive pathways from neurons to vessels are known to exist, whose individual effects added together sum to larger than observed changes in vasodilation (*Hosford and Gourine, 2019*), suggesting a non-linear interaction. It may be that for dilation occurring during sensory stimulation and sleep, the vasodilatory stimulus to vessels exceeds the saturation point, so that the loss of one vasodilatory pathway does not further affect the vascular response (*Figure 8*). This would also be consistent with isolated optogenetic or chemical stimulation

of type-I nNOS neurons being able to drive substantial dilation, while the loss of these neurons does not have a large impact on the vascular response.

Finally, pharmacological ablation and optogenetic/chemogenetic activation are not mirror manipulations and have differences that can produce non-symmetrical changes. Symmetrical changes would only occur if the neural circuitry and the signaling to the vessel were completely linear. Optogenetic activation/deactivation of a single-cell type does not produce symmetrical changes in the activity of other neurons (*Phillips and Hasenstaub, 2016*), so there is little reason to think ablation would have the exact opposite effect as activation. Furthermore, optogenetic/pharmacological activation of a single-cell type is unlikely to occur endogenously, as neural activity across different cell types is largely correlated (*Bugeon et al., 2022*). For example, during whisker stimulation, not only are type-I nNOS neurons in the somatosensory cortex activated (*Ruff et al., 2024*), but nearly every other cell type is as well (*Staiger and Petersen, 2021*), which means that type-I nNOS neurons are not the only neurons, and not even the only NO-producing neurons, sending signals to the vasculature during sensory stimulation.

One surprise is that we observed no changes in the vasodilation during NREM sleep, where type-I nNOS neurons are known to be active and play a role in inducing sleep (*Kilduff et al., 2011*). During NREM sleep, there is marked arterial dilation (*Gheres et al., 2023*; *Turner et al., 2020*), and it would seem natural that type-I nNOS neurons might drive this. However, we saw no difference in NREM dilation between the ablated and control mice. This could be due to compensation, saturation of the vasodilatory response, or it could be due to type-I nNOS neurons releasing other vasoconstrictory peptides (SST, NPY) so their net effects are canceled out. This last possibility is consistent with the loss of the post-stimulus vasoconstriction seen after type-I nNOS neuron ablation. Again, like with sensory stimulation, there may be a non-linearity in the vascular responses, so that another large factor, such as large neuromodulatory changes during sleep, could dominate over any other vascular signaling factor. The large norepinephrine decreases during NREM (*Kjaerby et al., 2022*; *Osorio-Forero et al., 2021*; *Turner et al., 2023*) could release the vessels from tonic vasoconstriction (as norepinephrine is a vasoconstrictor; *Bekar et al., 2012*), resulting in the dilation during NREM.

Finally, we saw a marked reduction in spontaneous vascular oscillation (vasomotion) at rest, both at the level of single arteries and at the level of blood volume. These spontaneous oscillations move cerebrospinal fluid (CSF) (*Holstein-Rønsbo et al., 2023*; *Kedarasetti et al., 2022*; *van Veluw et al., 2020*), which is important for clearing waste from the brain, and a reduction in amplitude will decrease the pumping efficacy. Additionally, there is a loss of coordination of neural and vascular dynamics across hemispheres after ablation of type-I nNOS neurons, indicated by the drop in coherence. Although a loss of synchronous dilations/constrictions might affect CSF movement, we might imagine that it would also adversely impact CSF pumping. Aged human brains show reduced interhemispheric synchrony in resting-state signals (*Zhao et al., 2020*), analogous to what we see here. type-I nNOS neurons also seem uniquely vulnerable to stress (*Han et al., 2019*), and loss of type-I nNOS neurons reduces power in the 1–4 Hz band of the LFP, which is positively associated with CSF clearance (*Hablitz et al., 2019*; *Jiang-Xie et al., 2024*). It is possible that adverse life experiences could cause the loss of type-I nNOS neurons, leading to a reduction in CSF flow seen in age (*Kress et al., 2014*; *Matrongolo et al., 2023*) that is thought to correlate with neurodegeneration.

## Materials and methods

This study was performed in accordance with the recommendations of the Guide for the Care and Use of Laboratory Animals of the National Institutes of Health. All procedures were performed in accordance with protocols approved by the Institutional Animal Care and Use Committee of Pennsylvania State University (Protocol 201042827). Data were acquired from 119 C57BL/6J mice (#000664, The Jackson Laboratory, Bar Harbor, ME) comprised of 57 males and 62 females between 3 and 9 months of age. Previous work from our lab has shown that the vasodilation elicited by whisker stimulation is the same in 2- to 4-month-old mice as in 18-month-old mice (*Bennett et al., 2024*). As the age range used here is spanned by this time interval, we would not expect any age-related differences. Food and water were provided ad libitum and animals were housed on a 12 hr light/dark cycle with all experiments occurring during the light cycle. Mice were individually housed after surgery. Sample sizes are consistent with previous studies (*Echagarruga et al., 2020*; *Turner et al., 2023*; *Turner et al., 2020*; *Zhang et al., 2021*) and based on a power analysis requiring 8–10 mice per group (Cohen's *d* = 1.3,

$\alpha = 0.05$, $(1 - \beta) = 0.800$). Experimenters were not blind to experimental conditions or data analysis except for histological experiments. Two SP-SAP mice were removed from the imaging datasets (24 SP-SAP remaining) due to not showing ablation of nNOS neurons during post-histological analysis, an attrition rate of approximately 8%.

## Surgical procedures

Saporin is a ribosome-inactivating protein that was conjugated to the Sar[9], Met(O$_2$)[11] analog of substance P (SP-SAP) or to a control peptide (Blank-SAP) (IT11 and IT21, Advanced Targeting Systems, Carlsbad, CA). Mice were anesthetized using 5% isoflurane (2% maintenance) vaporized in pure oxygen and were then injected intracortically with 4 ng of either SAP conjugate or Blank-SAP in 100 nl of artificial cerebrospinal fluid. The incision site was sterilized with betadine and 70% ethanol followed by a retraction of the skin atop the skull. A small (<0.5 mm) craniotomy was made above the vibrissae representation of somatosensory cortex (2 mm caudal, 3.25 mm lateral) and a sterile glass-pulled needle (tip diameter 50–100 µm) was inserted 500 µm beneath the cortical surface at 45°. The SAP conjugate was slowly injected at 100 nl/min using a programmable syringe pump (Harvard Apparatus, Holliston, MA) followed by closure of the incision with VetBond (3M, Saint Paul, MN). A subset of animals was also injected retro-orbitally with 25 µl of AAV PHP.eB-syn-jGCaMP7s-WPRE [$2 \times 10^{13}$ GC/ml] (104487-PHPeB, Addgene, Watertown, MA) diluted in 25 µl of sterile saline. Animals were given at least 2 weeks to recover prior to undergoing additional procedures. For imaging, a custom-machined titanium head bar was adhered atop the occipital bone of the skull using cyanoacrylate glue (Vibra-Tite 32402, ND Industries, Clawson, MI) and dental cement (Ortho-Jet, Lang Dental, Wheeling, IL). Self-tapping 3/32' #000 screws (J.I. Morris, Oxford, MA) were implanted in the frontal bones for structural stability. Electrodes were implanted into the cortex and hippocampus using paraformaldehyde (PFA)-coated tungsten stereotrodes (#795500, AM systems, Sequim, WA) for recordings of LFP and into the neck muscles using a pair of PFA-coated 7-strand stainless-steel wires (#793200, AM systems, Sequim, WA) for EMG. Polished and reinforced thinned-skull windows (*Drew et al., 2010*; *Shih et al., 2012*) were implanted over the somatosensory areas (either bilaterally or right hemisphere) using #0 coverslips (#72198, Electron Microscopy Sciences, Hatfield, PA). Detailed surgical procedures are as previously described (*Turner et al., 2020*).

## Data acquisition

Data were acquired with a custom LabVIEW program (National Instruments; https://github.com/DrewLab/LabVIEW-DAQ; *DrewLab, 2023*). Details on widefield optical imaging, two-photon microscopy, EMG, electrophysiology, vibrissae stimulation, and behavioral measurements including tracking of vibrissae and pupil diameter were performed as previously described (*Turner et al., 2023*; *Turner et al., 2020*; *Zhang et al., 2022*). Mice were gradually acclimated to head fixation of increasing duration (15, 30, and 60 min) on the days preceding the onset of experiments. The vibrissae (left, right, or a third air puffer not directed at the body as an auditory control) were randomly stimulated with air puffs [0.1 s or a train of 5-s pulses, 10 pounds force per square inch (PSI)] occurring every 30–45 s for the first 1 hr of imaging. Data were acquired in 15-min intervals with a brief (~30 s) gap in between for saving data to disk. Each animal was run for up to 6 imaging sessions lasting from 1 to 5 hr depending on experiment. In two-photon experiments, mice were briefly anesthetized and retro-orbitally injected with 100 µl of 5% (wt/vol) fluorescein isothiocyanate–dextran (FITC) (FD150S, Sigma-Aldrich, St. Louis, MO) dissolved in sterile saline. Imaging was done on a Sutter Movable Objective Microscope with a CFI75 LWD 16X W objective (Nikon, Melville, NY) and a MaiTai HP Ti:sapphire laser (Spectra-Physics, Santa Clara, CA) tuned to 800 nm.

## Histology

Following the conclusion of imaging experiments, animals were deeply anesthetized and transcardially perfused with heparin-saline followed by 4% PFA. Presence or absence of nNOS-positive neurons in the injected hemisphere was verified using nicotinamide adenine dinucleotide phosphate (NADPH)-diaphorase staining for localizing the sparsely populated type-I nNOS neurons (*Scherer-Singler et al., 1983*).

## Immunohistochemistry

All histological analyses were done blinded to the experimental condition. Mice were deeply anesthetized with 5% isoflurane and perfused transcardially with ice-cold phosphate-buffered saline (PBS, pH 7.4) and 4% PFA (pH 7.4). Brains were removed, post-fixed in PFA for 24 hr and stored in PBS at 4°C for less than 1 week. A fiduciary mark was placed in the right hemisphere. 40 μm free-floating sections were sliced with a Leica vibratome (VS 1200, Leica) and stored in PBS for less than 1 week. Prior to immunostaining, slices were washed three times in PBS for 10 min each and underwent antigen retrieval in 10 mM sodium citrate buffer (pH 6.0) at 80°C for 30 min. Slices were washed three times in PBS for 10 min each and permeabilized in 0.5% Triton X-100 in PBS for 60 min. Nonspecific binding was blocked with 5% normal donkey serum (NDS) (ab7475) in 0.1% Triton X-100 in PBS for 60 min. Slices were then incubated in a primary antibody cocktail, including goat anti-nNOS (1:500, ab1376), rabbit anti-IBA-1 (1:500, ab178847), rat anti-GFAP (1:500, Thermo Fisher 13-0300), rabbit anti-NeuN (1:2000, EnCor), rabbit anti-TACR1 (1:500 Invitrogen PA1-16713) in 2.5% NDS in 0.1% Triton X-100 in PBS for 48 hr at 4°C. Slices were rinsed three times with PBS for 10 min each and incubated in a fluorophore-tagged secondary antibody cocktail, including donkey anti-rabbit Alexa Fluor 488 (1:500, ab150073), donkey anti-goat Alexa Fluor 594 (1:500, ab150132), donkey anti-rabbit Alexa Fluor 647 (1:500, ab150075), donkey anti-rat Alexa Fluor 488 (1:500 ab150153), for 4 hr at room temperature. Slices were rinsed again three times with PBS, with the last step including DAPI (1:10,000, 10 mg/ml, Millpore Sigma, 10236276001), mounted on glass slides, air-dried, and cover slipped with Immunomount (Thermo Fisher Scientific, Waltham, MA, United States). Images were obtained with an Olympus BX63 upright microscope (Center Valley, PA, United States) under matched exposure settings. Three to eight images from both hemispheres were taken per region.

## Cell counting and immunofluorescence quantification

Total cell counts and absolute changes in immunofluorescence (*Figure 1*) were quantified using ImageJ (National Institutes of Health, Bethesda, MD, United States). The region of interest (ROI) for analysis of cell counts was determined based on the injection site for both SP-SAP and Blank SAP injections, with a 1-mm diameter circle centered around the injection site and averaged across three to five sections where available. In most animals, the SP-SAP had a lateral spread greater than 500 μm and encompassed the entire depth of cortex (1–1.5 mm in SI). For total cell counts, an ROI was delineated, and cells were automatically quantified under matched criteria for size, circularity, and intensity. Image threshold was adjusted until absolute value percentages were between 1–10% of the histogram density. The function *Analyze Particles* was then used to estimate the number of particles with a size of 100–99,999 pixels$^2$ and a circularity between 0.3 and 1.0 (*Dao et al., 2020*; *Sicher et al., 2023*; *Smith et al., 2020*). Immunoreactivity was quantified as mean fluorescence intensity of the ROI (*Pleil et al., 2015*).

## Data analysis

Data were analyzed with code (MATLAB 2019b-2024a, MathWorks).

## Statistical analysis

Statistical evaluations were made using either GLME, unpaired *t*-test, or GLM. GLME models had the arousal state as a fixed effect, mouse identity as a random effect, and hemisphere [left/right (L/R), if applicable] as an interaction with the animal ID, or using a paired *t*-test where appropriate. Unless otherwise stated, statistical results report p values from a GLME test. All reported quantifications are mean ± SD unless otherwise indicated. Unless otherwise noted, all pupil diameter measurements are in z-units. MATLAB functions used were fitglme, ttest, fitglm.

## Hemodynamic correction

Widefield imaging was done with a Dalsa 1M60 Pantera CCD camera (Phase One, Cambridge, MA) with a magnifying lens (VZM 300i, Edmund Optics, Barrington, NJ). Reflectance measurements (*Figure 2*) were converted to changes in total hemoglobin (Δ[HbT]), oxy-hemoglobin (Δ[HbO]), and deoxy-hemoglobin (Δ[HbR]) using the Beer–Lambert law (*Ma et al., 2016a*; *Ma et al., 2016b*). Correction for attenuation of GCaMP7s fluorescence due to absorption of the surrounding tissue was corrected as previously described (*Kramer and Pearlstein, 1979*; *Ma et al., 2016a*; *Scott et al.,*

*2018*; *Wright et al., 2017*). Changes in fluorescence intensity in the GCaMP7s signal due to blood absorption were approximated in a pixel-wise fashion by multiplying each value by the ratio of the green and blue channel's resting baseline pixel value.

## Two-photon image processing

Image stacks were corrected for $x$–$y$ motion artifacts and aligned with a rigid registration algorithm. A rectangular box was drawn around a straight, evenly illuminated vessel segment, and the pixel intensity was averaged along the long axis to calculate the vessel's diameter from the full-width at half-maximum (https://github.com/DrewLab/Surface-Vessel-FWHM-Diameter; *DrewLab, 2024*; *Drew et al., 2011*). The diameter of penetrating arterioles was calculated using the thresholding in Radon space (TiRS) algorithm (https://github.com/DrewLab/Thresholding_in_Radon_Space; *DrewLab, 2020*; *Gao and Drew, 2014*; *Gao et al., 2015*).

## Electrophysiology

Gamma-band [30–100 Hz] LFP band was digitally bandpass filtered from recorded broadband data using a third-order Butterworth filter, squared, low-pass filtered below 1 Hz, and resampled at 30 Hz. Time–frequency spectrograms (*Figure 2*) were calculated using the Chronux toolbox version 2.12 v03 (*Bokil et al., 2010*), function mtspecgramc with a 5-s window and 1/5 s step size using [5,9] tapers and a passband of 1–100 Hz to encompass the LFP. EMG (300–3 kHz) from the neck muscles was bandpass filtered, squared, convolved with a Gaussian kernel with a standard deviation of 0.5 s, log transformed, then resampled at 30 Hz. MATLAB function(s): butter, zp2sos, filtfilt, gausswin, log10, conv, resample.

## Evoked responses and slope

Evoked responses (*Figure 3*) including whisker stimulation and locomotion for the various data types ([HbT], [HbO], [HbR], GCaMP7s, arteriole diameter) were compared between the indicated intervals (i.e., 2:4 s post-stimulation). The average of the 2 s preceding event onsets was subtracted from the event and smoothed with a third-order Savitzky-Golay filter. The average slope was calculated by comparing the rise (Δ[HbT]) over the run (neural data) before being fit with a linear model forced through the origin. MATLAB function(s): sgolayfilt.

## Spectral power and coherence

Spectral power (*Figures 4–6*, *Figure 5—figure supplement 1*) was estimated using the Chronux toolbox (*Bokil et al., 2010*) function mtspectrumc. Data was detrended within individual events and truncated to the desired length depending on behavior (10 s for rest, 15 min for Alert and Asleep). Coherence analysis between two signals within the same hemisphere or between signals recorded bilaterally was run for each data type using the Chronux function coherency after detrending using the MATLAB function detrend.

## Cross-correlation

Cross-correlations (*Figure 5*) between gamma-band power or $\Delta F/F$ and changes in total hemoglobin Δ[HbT] were taken during periods of resting-state, alert, and asleep. Data were mean-subtracted and digitally low-pass filtered (<1 Hz) with a fourth-order Butterworth filter (MATLAB function(s): butter, zp2sos, filtfilt). Cross-correlation analysis was run for each arousal state (MATLAB function(s): xcorr) with a ± 5 s lag time.

## Resting-state variance and Δ[HbT] during different arousal states

Variance during the resting state for both Δ[HbT] and diameter signals (*Figure 7*) was taken from resting-state events lasting ≥10 s in duration. Average Δ[HbT] from within the 1 mm ROI over the vibrissae representation of SI during each arousal state was taken with respect to awake resting baseline events ≥10 s in duration. Continuous NREM sleep events ≥30 s, REM sleep events ≥60 s, and periods following administration of isoflurane were compared between groups. Each event was digitally low-pass filtered (<1 Hz) with a fourth-order Butterworth filter (MATLAB function(s): butter, zp2sos, filtfilt) and then averaged within each individual time series prior to comparing across animals/groups.

## Open field behavior

Exploratory behavior (*Figure 1—figure supplement 2*) was measured in a custom-made 30 cm × 60 cm arena during the light phase. Mice were not habituated to the arena. Mice were placed in the arena for 10 min to explore, and their behavior was recorded using an overhead camera (Blackfly BFLY-U3-23S6M, Teledyne FLIR, Wilsonville, OR) at a frame rate of either 15 or 30 frames per second. The arena was illuminated with 780 nm light. The surroundings of the arena were dark during the entire recording session. The arena was cleaned with 70% ethanol in between animals. Only the first 5 min of the behavior were analyzed and reported here. The total distance traveled over the first 5 min, and the time spent in the 25 × 55 cm center rectangle was quantified. GLME was used in MATLAB to perform statistical analysis. No statistical outliers were removed from the data. Mouse behavior was tracked using DeepLabCut (*Mathis et al., 2018*) and analyzed with custom MATLAB algorithms (*Brockway et al., 2023*; *Zhang et al., 2022*). Eight points on the body (left ear, right ear, head, mid, back near the hip joint, base of the tail, midpoint of the tail, and end of the tail) were tracked. Four markers were placed in the arena's four corners with an additional four points calculated from the four corners to track the center rectangle. A DeepLabCut model was trained and evaluated on a subset of mice before applying the model to all the mice. DeepLabCut tracking was considered acceptable if the tracking confidence was above 97%; however, in most cases, it was higher than 99%. Tracking positions were exported to a CSV file containing the tracked location's *XY* coordinates (frame pixels). Tracked videos from random mice were visualized to confirm the accuracy of the tracking. The experimenter was blinded to drug injection group identification until tracking and analysis were completed. The distance was calculated from the first 5 min as Euclidean distance between the point tracked in the middle of the mouse body from subsequent frames. Center time was quantified as the number of frames a mouse spends within the polygon bounded by four points tracked on the center of the arena.

## Sleep scoring

Sleep scoring (*Figure 1—figure supplement 3*) was performed as in *Turner et al., 2020*. Briefly, arousal state was scored using a combination of cortical and hippocampal LFP, EMG, pupil diameter, and whisker and body movement. Periods of manually chosen awake rest 5 s in duration with no vibrissae or body motion were used as a baseline to normalize neural and hemodynamic signals. NREM sleep shows elevated cortical delta-band power and lower EMG power. REM sleep is marked by elevated hippocampal theta-band power and elevated cortical gamma-band power with very low baseline EMG power (muscle atonia; *Cantero et al., 2004*; *Montgomery et al., 2008*; *Le Van Quyen et al., 2010*; *Sullivan et al., 2014*). Every 5-s interval was classified as either Awake, NREM sleep, or REM sleep using a bootstrap aggregating random forest model with the predictors of cortical delta-band power, cortical beta band power, cortical gamma LFP, hippocampal theta LFP, EMG power, heart rate, and whisking duration. Sleep model accuracy was validated using the out-of-bag error during model training. MATLAB functions used were TreeBagger, oobError, predict.

## Pupil diameter and interblink interval

Pupil diameter (*Figure 1—figure supplement 2*) was low-pass filtered at 1 Hz with a fourth-order Butterworth filter. Changes in whisking-evoked and stimulus-evoked (contralateral, auditory) diameters normalized relative to the mean of the 2 s preceding the event onset. MATLAB functions used were butter, zp2sos, and filtfilt. Interblink interval and blink-associated physiology. Because of breaks in recording to save the data to disk, interblink interval was calculated between blinks occurring within 15-min records and not blinks on the edges of trials. Blinks that occurred within 1 s of each other were linked together as blinking bouts, and all blink-triggered analyses were with respect to the first blink in a series. Blink-triggered averages were separated into two groups depending on the arousal state classification of the 5-s bin before the blink, being either Awake (arousal state classification of Awake) or Asleep (arousal state classification being either NREM or REM).

## Hemodynamic response function

The HRF (*Figure 5—figure supplement 1*) was calculated as previously described (*Winder et al., 2017*; *Zhang et al., 2023*) using both deconvolution and fitting the response to a gamma distribution. Only neural activity (gamma-band power, 30–100 Hz) within 1.5 s of the stimulus was used for calculating the HRF. Each HRF was calculated using half of the data and tested on the other half. To test

the predictive capability of each HRF, the impulse function was fit with a gamma-distribution function and convolved with the neural power to predict the measured hemodynamic signal. The coefficient of determination ($R^2$) between the predicted and actual hemodynamic signal was evaluated to quantify the efficacy of HRF's prediction (MATLAB function(s): sgolayfilt, detrend, fminsearch, fitlm).

## Pearson's correlation coefficient calculations

The Pearson's correlation coefficient between bilateral signals (Figure S6) was obtained by mean-subtracting and digitally low-pass filtering (<1 Hz) with a fourth-order Butterworth filter and then taking the Pearson's correlation coefficient (MATLAB function(s): butter, zp2sos, filtfilt, corrcoef).

## Arousal state transitions

Transitions between arousal states (Awake to NREM, NREM to Awaken, NREM to REM, REM to Awake) were compared across groups (*Figure 7—figure supplement 1*) by averaging events that met the criteria of 30 s of one state's consecutive classifications followed by 30 s of another state. The difference in the Δ[HbT] between the pre- and post-state transition was compared by taking the 30:10 s prior to the transition zero point to the 10:30 s following. [HbT] was digitally low-pass (<1 Hz) filtered using a fourth-order Butterworth filter (MATLAB function(s): butter, zp2sos, filtfilt).

## Acknowledgements

This work was supported by NIH Grants R01NS078168, R01NS079737, U19NS128613 to PJD, R01AA029403 to NAC, an American Heart Association Predoctoral fellowship 24PRE1201066 to MSH and F31ES036154 to DIG. DIG was supported by T32GM108563 and MSH was supported by T32NS115667.

## Additional information

### Funding

| Funder | Grant reference number | Author |
| --- | --- | --- |
| National Institute of Neurological Disorders and Stroke | R01NS078168 | Patrick J Drew |
| National Institute of Neurological Disorders and Stroke | R01NS079737 | Patrick J Drew |
| National Institute of Neurological Disorders and Stroke | U19NS128613 | Patrick J Drew |
| National Institute on Alcohol Abuse and Alcoholism | R01AA029403 | Nicole Crowley |
| American Heart Association | 10.58275/aha.24pre1201066.pc.gr.190598 | Md Shakhawat Hossain |
| National Institute of Environmental Health Sciences | F31ES036154 | Denver Greenawalt |
| National Institute of General Medical Sciences | T32GM108563 | Denver Greenawalt |
| National Institute of Neurological Disorders and Stroke | T32NS115667 | Md Shakhawat Hossain |
| American Heart Association | 935961 | Qingguang Zhang |

| Funder | Grant reference number | Author |
|---|---|---|

The funders had no role in study design, data collection, and interpretation, or the decision to submit the work for publication.

## Author contributions

Kevin Turner, Conceptualization, Data curation, Software, Formal analysis, Investigation, Visualization, Methodology, Writing – original draft, Writing – review and editing; Dakota Brockway, Data curation, Formal analysis, Investigation, Visualization; Md Shakhawat Hossain, Data curation, Software, Funding acquisition, Investigation, Visualization, Methodology; Keith Griffith, Investigation; Denver Green-awalt, Funding acquisition, Investigation, Visualization, Methodology; Qingguang Zhang, Software, Funding acquisition, Investigation, Visualization, Methodology; Kyle Gheres, Software, Investigation, Methodology; Nicole Crowley, Conceptualization, Supervision, Funding acquisition, Writing – original draft, Project administration, Writing – review and editing; Patrick J Drew, Conceptualization, Formal analysis, Supervision, Funding acquisition, Writing – original draft, Project administration, Writing – review and editing

## Author ORCIDs

Kevin Turner ⓘ https://orcid.org/0000-0002-3044-7079
Dakota Brockway ⓘ https://orcid.org/0009-0003-5392-9630
Md Shakhawat Hossain ⓘ https://orcid.org/0000-0001-9897-0854
Qingguang Zhang ⓘ https://orcid.org/0000-0003-4500-813X
Kyle Gheres ⓘ https://orcid.org/0000-0001-7568-9023
Nicole Crowley ⓘ https://orcid.org/0000-0002-1142-7929
Patrick J Drew ⓘ https://orcid.org/0000-0002-7483-7378

## Ethics

This study was performed in accordance with the recommendations of the Guide for the Care and Use of Laboratory Animals of the National Institutes of Health. All procedures were performed in accordance with protocols approved by the Institutional Animal Care and Use Committee of Pennsylvania State University (Protocol 201042827). All surgery was performed under isoflurane anesthesia, and every effort was made to minimize suffering.

Reviewer #1 (Public review): https://doi.org/10.7554/eLife.105649.3.sa1
Reviewer #2 (Public review): https://doi.org/10.7554/eLife.105649.3.sa2
Author response https://doi.org/10.7554/eLife.105649.3.sa3

# Additional files

## Supplementary files

MDAR checklist

## Data availability

The data and code for generating the figures are available at Dryad and analysis code is available at GitHub (copy archived at *Turner, 2025*).

The following dataset was generated:

| Author(s) | Year | Dataset title | Dataset URL | Database and Identifier |
|---|---|---|---|---|
| Turner K, Brockway D, Hossain M, Griffith K, Greenawalt D, Zhang Q, Gheres K, Crowley N, Drew PJ | 2025 | Type-I nNOS neurons orchestrate cortical neural activity and vasomotion | https://doi.org/10.5061/dryad.tb2rbp0bq | Dryad Digital Repository, 10.5061/dryad.tb2rbp0bq |

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
