## [Editor Report · eLife Assessment]

This **important** study provides **solid** evidence for new insights into the role of Type-1 nNOS interneurons in driving neuronal network activity and controlling vascular network dynamics in awake, head-fixed mice. The authors use an original strategy based on the ablation of Type-1 nNOS interneurons with local injection of saporin conjugated to a substance P analogue into the somatosensory cortex. They show that ablation of type I nNOS neurons has surprisingly little effect on neurovascular coupling, although it alters neural activity and vascular dynamics.

---

## [Referee Report · Reviewer #1 (Public review)]

Turner et al. present an original approach to investigate the role of Type-1 nNOS interneurons in driving neuronal network activity and in controlling vascular network dynamics in awake head-fixed mice. Selective activation or suppression of Type-1 nNOS interneurons has previously been achieved using either chemogenetic, optogenetic or local pharmacology. Here, the authors took advantage of the fact that Type-1 nNOS interneurons are the only cortical cells that express the tachykinin receptor 1 to ablate them with a local injection of saporin conjugated to substance P (SP-SAP). SP-SAP causes cell death in 90 % of type1 nNOS interneurons without affecting microglia, astrocytes and neurons. The authors report that the ablation has no major effects on sleep or behavior. Refining the analysis by scoring neural and hemodynamic signals with electrode recordings, calcium signal imaging and wide field optical imaging, they observe that Type-1 nNOS interneuron ablation does not change the various phases of the sleep/wake cycle. However, it does reduce low-frequency neural activity, irrespective of the classification of arousal state. Analyzing neurovascular coupling using multiple approaches, they report small changes in resting-state neural-hemodynamic correlations across arousal states, primarily mediated by changes in neural activity. Finally, they show that nNOS type 1 interneurons play a role in controlling interhemispheric coherence and vasomotion.

In conclusion, these results are interesting, use state-of-the-art methods and are well supported by the data and their analysis. I have only a few comments on the stimulus-evoked haemodynamic responses that can be easily addressed:

Comments on revisions:

As I mentioned in my initial review, this study is important. In my opinion, it could be published as is. Nonetheless, I am still somewhat dissatisfied with the authors' responses to my earlier comments. I understand that the same animals were not used for both stimulation paradigms, which is unfortunate. Nonetheless, I would have appreciated it if the authors had provided a couple of experiments illustrating GCaMP7 signals during brief stimulation in their reply to the reviewers. I am still unconvinced by the authors' suggestion that the GCaMP7 signal would remain stable during removal of the vascular undershoot. Since the absence of the undershoot is notable, I anticipate that a significant part of the initial response to prolonged stimulation is influenced by processes that occur during the 0.1-second stimulation, processes that may involve a change in the bulk neuronal response.

In short, the data could support or refute the following statement: "Loss of type-I nNOS neurons drove minimal changes in the vasodilation elicited by brief stimulation..."

---

## [Referee Report · Reviewer #2 (Public review)]

Summary:

This important study by Turner et al., examines the functional role of a sparse but unique population of neurons in the cortex that express Nitric oxide synthase (Nos1). To do this, they pharmacolologically ablate these neurons in focal region of whisker related primary somatosensory (S1) cortex using a saponin-Substance P conjugate. Using widefield and 2-photon microscopy, as well as field recordings, they examine the impact of this cell specific lesion on blood flow dynamics and neuronal population activity. Within primary somatosensory cortex after Nos1 ablation, they find changes in neural activity patterns, decreased delta band power, reduced sensory evoked changes in blood flow (specifically eliminates the sustained blood flow change after stimulation) and decreased vasomotion.

Strengths:

This was a technically challenging study and the experiments were executed in an expert manner. The manuscript was well written and I appreciated the cartoon summary diagrams included in each figure. The analysis was rigorous and appropriate. Their discovery that Nos1 neurons can have significant effects on blood flow dynamics and neural activity is quite novel that should seed many follow up, mechanistic experiments to explain this phenomenon. The conclusions were justified by the convincing data presented.

Weaknesses:

I did not find any major flaws with the study. I originally noted some potential issues with the authors' characterization of the lesion and its extent, but that has been resolved in the revised manuscript.

Comments on revisions:

The authors have thoughtfully addressed the relatively minor concerns I had originally raised. Congratulations to the authors for producing this important paper.

---

## [Author Response]

The following is the authors’ response to the original reviews.

**Reviewer #1 (Public review):**
Turner et al. present an original approach to investigate the role of Type-1 nNOS interneurons in driving neuronal network activity and in controlling vascular network dynamics in awake head-fixed mice. Selective activation or suppression of Type-1 nNOS interneurons has previously been achieved using either chemogenetic, optogenetic, or local pharmacology. Here, the authors took advantage of the fact that Type-1 nNOS interneurons are the only cortical cells that express the tachykinin receptor 1 to ablate them with a local injection of saporin conjugated to substance P (SP-SAP). SP-SAP causes cell death in 90 % of type1 nNOS interneurons without affecting microglia, astrocytes, and neurons. The authors report that the ablation has no major effects on sleep or behavior. Refining the analysis by scoring neural and hemodynamic signals with electrode recordings, calcium signal imaging, and wide-field optical imaging, the authors observe that Type-1 nNOS interneuron ablation does not change the various phases of the sleep/wake cycle. However, it does reduce low-frequency neural activity, irrespective of the classification of arousal state. Analyzing neurovascular coupling using multiple approaches, they report small changes in resting-state neural-hemodynamic correlations across arousal states, primarily mediated by changes in neural activity. Finally, they show that nNOS type 1 interneurons play a role in controlling interhemispheric coherence and vasomotion.In conclusion, these results are interesting, use state-of-the-art methods, and are well supported by the data and their analysis. I have only a few comments on the stimulus-evoked haemodynamic responses, and these can be easily addressed.

We thank the reviewer for their positive comments on our work.

**Reviewer #2 (Public review):**
Summary:This important study by Turner et al. examines the functional role of a sparse but unique population of neurons in the cortex that express Nitric oxide synthase (Nos1). To do this, they pharmacologically ablate these neurons in the focal region of whisker-related primary somatosensory (S1) cortex using a saponin-substance P conjugate. Using widefield and 2photon microscopy, as well as field recordings, they examine the impact of this cell-specific lesion on blood flow dynamics and neuronal population activity. Locally within the S1 cortex, they find changes in neural activity paFerns, decreased delta band power, and reduced sensory-evoked changes in blood flow (specifically eliminating the sustained blood flow change amer stimulation). Surprisingly, given the tiny fraction of cortical neurons removed by the lesion, they also find far-reaching effects on neural activity paFerns and blood volume oscillations between the cerebral hemispheres.Strengths:This was a technically challenging study and the experiments were executed in an expert manner. The manuscript was well wriFen and I appreciated the cartoon summary diagrams included in each figure. The analysis was rigorous and appropriate. Their discovery that Nos1 neurons can have far-reaching effects on blood flow dynamics and neural activity is quite novel and surprising (to me at least) and should seed many follow-up, mechanistic experiments to explain this phenomenon. The conclusions were justified by the convincing data presented.Weaknesses:I did not find any major flaws in the study. I have noted some potential issues with the authors' characterization of the lesion and its extent. The authors may want to re-analyse some of their data to further strengthen their conclusions. Lastly, some methodological information was missing, which should be addressed.

We thank the reviewer for their enthusiasm for our work.

**Reviewer #3 (Public review):**
The role of type-I nNOS neurons is not fully understood. The data presented in this paper addresses this gap through optical and electrophysiological recordings in adult mice (awake and asleep).This manuscript reports on a study on type-I nNOS neurons in the somatosensory cortex of adult mice, from 3 to 9 months of age. Most data were acquired using a combination of IOS and electrophysiological recordings in awake and asleep mice. Pharmacological ablation of the type-I nNOS populations of cells led to decreased coherence in gamma band coupling between lem and right hemispheres; decreased ultra-low frequency coupling between blood volume in each hemisphere; decreased (superficial) vascular responses to sustained sensory stimulus and abolishment of the post-stimulus CBV undershoot. While the findings shed new light on the role of type-I nNOS neurons, the etiology of the discrepancies between current observations and literature observations is not clear and many potential explanations are put forth in the discussion.

We thank the reviewer for their comments.

**Reviewer #1 (Recommendations for the authors):**
(1) Figure 3, Type-1 nNOS interneuron ablation has complex effects on neural and vascular responses to brief (.1s) and prolonged (5s) whisker stimulation. During 0.1 s stimulation, ablation of type 1 nNOS cells does not affect the early HbT response but only reduces the undershoot. What is the pan-neuronal calcium response? Is the peak enhanced, as might be expected from the removal of inhibition? The authors need to show the GCaMP7 trace obtained during this short stimulation.

Unfortunately, we did not perform brief stimulation experiments in GCaMP-expressing mice. As we did not see a clear difference in the amplitude of the stimulus-evoked response with our initial electrophysiology recordings (Fig. 3a), we suspected that an effect might be visible with longer duration stimuli and thus pivoted to a pulsed stimulation over the course of 5 seconds for the remaining cohorts. It would have been beneficial to interweave short-stimulus trials for a direct comparison between the complimentary experiments, but we did not do this.

During 5s stimulation, both the early and delayed calcium/vascular responses are reduced. Could the authors elaborate on this? Does this mean that increasing the duration of stimulation triggers one or more additional phenomena that are sensitive to the ablation of type 1 nNOS cells and mask what is triggered by the short stimulation? Are astrocytes involved? How do they interpret the early decrease in neuronal calcium?

As our findings show that ablation reduces the calcium/vascular response more prominently during prolonged stimulation, we do suspect that this is due to additional NO-dependent mechanisms or downstream responses. NO is modulator of neural activity, generally increasing excitability (Kara and Friedlander 1999, Smith and Otis 2003), so any manipulation that changes NO levels will change (likely decrease) the excitability of the network, potentially resulting in a smaller hemodynamic response to sensory stimulation secondary to this decrease. While short stimuli engage rapid neurovascular coupling mechanisms, longer duration (>1s) stimulation could introduce additional regulatory elements, such as astrocytes, that operate on a slower time scale. On the right, we show a comparison of the control groups ploFed together from Fig. 3a and 3b with vertical bars aligned to the peak. During the 5s stimulation, the time-to-peak is roughly 830 milliseconds later than the 0.1s stimulation, meaning it’s plausible that the signals don’t separate until later. Our interpretation is that the NVC mechanisms responsible for brief stimulus-evoked change are either NO-independent or are compensated for in the SSP-SAP group by other means due to the chronic nature of the ablation.

We have added the following text to the Discussion (Line 368): “Loss of type-I nNOS neurons drove minimal changes in the vasodilation elicited by brief stimulation, but led to decreased vascular responses to sustained stimulation, suggesting that the early phase of neurovascular coupling is not mediated by these cells, consistent with the multiple known mechanisms for neurovascular coupling (AFwell et al 2010, Drew 2019, Hosford & Gourine 2019) acting through both neurons and astrocytes with multiple timescales (Le Gac et al 2025, Renden et al 2024, Schulz et al 2012, Tran et al 2018).”

(2) In Figures 4d and e, it is unclear to me why the authors use brief stimulation to analyze the relationship between HbT and neuronal activity (gamma power) and prolonged stimulation for the relationship between HbT and GCaMP7 signal. Could they compare the curves with both types of stimulation?

As discussed previously, we did not use the same stimulation parameters across cohorts. The mice with implanted electrodes received only brief stimulation, while those undergoing calcium imaging received longer duration stimulus.

**Reviewer #2 (Recommendations for the authors):**
(1) Results, how far-reaching is the cell-specific ablation? Would it be possible to estimate the volume of the cortex where Nos1 cells are depleted based on histology? Were there signs of neuronal injury more remotely, for example, beading of dendrites?

We regularly see 1-2 mm in diameter of cell ablation within the somatosensory cortex of each animal, which is consistent with the spread of small molecules. Ribosome inactivating proteins like SAP are smaller than AAVs (~5 nm compared to ~25 nm in diameter) and thus diffuse slightly further. We observed no obvious indication of neuronal injury more remotely or in other brain regions, but we did not image or characterize dendritic beading, as this would require a sparse labeling of neurons to clearly see dendrites (NeuN only stains the cell body). Our histology shows no change in cell numbers.

We have added the following text to the Results (Line 124): “Immunofluorescent labeling in mice injected with Blank-SAP showed labeling of nNOS-positive neurons near the injection site. In contrast, mice injected with SP-SAP showed a clear loss in nNOS-labeling, with a typical spread of 1-2 mm from the injection site, though nNOS-positive neurons both subcortically and in the entirety of the contralateral hemisphere remaining intact.”

(2) For histological analysis of cell counts amer the lesion, more information is needed. How was the region of interest for counting cells determined (eg. 500um radius from needle/pipeFe tract?) and of what volume was analysed?

The region of interest for both SSP-SAP and Blank SAP injections was a 1 mm diameter circle centered around the injection site and averaged across sections (typically 3-5 when available). In most animals, the SSP-SAP had a lateral spread greater than 500 microns and encompassed the entire depth of cortex (1-1.5 mm in SI, decreasing in the rostral to caudal direction). The counts within the 1 mm diameter ROI were averaged across sections and then converted into the cells per mm area as presented. Note the consistent decrease in type I nNOS cells seen across mice in Fig 1d, Fig S1b.

We have added the following text in the Materials & Methods (Line 507): “The region of interest for analysis of cell counts was determined based on the injection site for both SP-SAP and Blank SAP injections, with a 1 mm diameter circle centered around the injection site and averaged across 3-5 sections where available. In most animals, the SP-SAP had a lateral spread greater than 500 microns and encompassed the entire depth of cortex (1-1.5 mm in SI).”

(3) Based on Supplementary Figure 1, it appears that the Saponin conjugate not only depletes Nos neurons but also may affect vascular (endothelial perhaps) Nos expression. Some quantification of this effect and its extent may be insighIul in terms of ascribing the effects of the lesion directly on neurons vs indirectly and perhaps more far-reaching via vascular/endothelial NOS.

Thank you for this comment. While this is a possibility, while we have found that the high nNOS expression of type-I nnoos neurons makes NADPH diaphorase a good stain for detecting them, it is less useful for cell types that expres NOS at lower levels. We have found that the absolute intensity of NADPH diaphorase staining is somewhat variable from section to section. Variability in overall NADPH diaphorase intensity is likely due to several factors, such as duration of staining, thickness of the section, and differences in PFA concentration within the tissue and between animals. As NADPH diaphorase staining is highly sensitive to amount PFA exposure, any small differences in processing could affect the intensity, and slight differences in perfusion quality and processing could account. A second, perhaps larger issue could be due to differences in the number of arteries (which will express NOS at much higher levels than veins, and thus will appear darker) in the section. We did not stain for smooth muscle and so cannot differentiate arteries and veins. Any difference in vessel intensity could be due to random variations in the numbers of arteries/veins in the section. While we believe that this is a potentially interesting question, our histological experiments were not able to address it.

(4) The assessment for inflammation took place 1 month amer the lesion, but the imaging presumably occurred ~ 2 weeks amer the lesion. Note that it seemed somewhat ambiguous as to when approximately, the imaging, and electrophysiology experiments took place relative to the induction of the lesion. Presumably, some aspects of inflammation and disruption could have been missed, at the time when experiments were conducted, based on this disparity in assessment. The authors may want to raise this as a possible limitation.

We apologize for our unclear description of the timeline. We began imaging experiments at least 4 weeks amer ablation, the same time frame as when we performed our histological assays.

We have added the following text to the Discussion (Line 379): “With imaging beginning four weeks amer ablation, there could be compensatory rewiring of local and/or network activity following type-I nNOS ablation, where other signaling pathways from the neurons to the vasculature become strengthened to compensate for the loss of vasodilatory signaling from the typeI nNOS neurons.”

(5) Results Figure 2, please define "P or delta P/P". Also, for Figure 2c-f, what do the black vertical ticks represent?

∆P/P is the change in the gamma-band power relative to the resting-state baseline, and black tick marks indicate binarized periods of vibrissae motion (‘whisking’). We have clarified this in Figure caption 2 (Line 174).

(6) Figure 3b-e, is there not an undershoot (eventually) amer 5s of stimulation that could be assessed?

Previous work has shown that there is no undershoot in response to whisker stimulations of a few seconds (Drew, Shih, Kelinfeld, PNAS, 2011). The undershoot for brief stimuli happens within ~2.5 s of the onset/cessation of the brief stimulation, this is clearly lacking in the response to the 5s stim (Fig 3). The neurovascular coupling mechanisms recruited during the short stimulation are different than those recruited during the long stimulus, making a comparison of the undershoot between the two stimulation durations problematic.

For Figures 3e and 6 how was surface arteriole diameter or vessel tone measured? 2P imaging of fluorescent dextran in plasma? Please add the experimental details of 2P imaging to the methods. Including some 2P images in the figures couldn't hurt to help the reader understand how these data were generated.

We have added details about our 2-photon imaging (FITC-dextran, full-width at half-maximum calculation for vessel diameter) as well as a trace and vessel image to Figure 2.

We have added the following text to the Materials & Methods (Line 477): “In two-photon experiments, mice were briefly anesthetized and retro-orbitally injected with 100 µL of 5% (weight/volume) fluorescein isothiocyanate–dextran (FITC) (FD150S, Sigma-Aldrich, St. Louis, MO) dissolved in sterile saline.”

We have added the following text to the Materials & Methods (Line 532): “A rectangular box was drawn around a straight, evenly-illuminated vessel segment and the pixel intensity was averaged along the long axis to calculate the vessel’s diameter from the full-width at half-maximum (https://github.com/DrewLab/Surface-Vessel-FWHM-Diameter; (Drew, Shih et al. 2011)).”

(7) Did the authors try stimulating other body parts (eg. limb) to estimate how specific the effects were, regionally? This is more of a curiosity question that the authors could comment on, I am not recommending new experiments.

We did measure changes in [HbT] in the FL/HL representation of SI during locomotion (Line 205), which is known to increase neural activity in the somatosensory cortex (Huo, Smith and Drew, Journal of Neuroscience, 2014; Zhang et al., Nature Communications 2019). We observed a similar but not statistically significant trend of decreased [HbT] in SP-SAP compared to control. This may have been due to the sphere of influence of the ablation being centered on the vibrissae representation and not having fully encompassed the limb representation. We agree with the referee that it would be interesting to characterize these effects on other sensory regions as well as brain regions associated with tasks such as learning and behavior.

(8) Regarding vasomotion experiments, are there no other components of this waveform that could be quantified beyond just variance? Amplitude, frequency? Maybe these don't add much but would be nice to see actual traces of the diameter fluctuations. Further, where exactly were widefield-based measures of vasomotion derived from? From some seed pixel or ~1mm ROI in the center of the whisker barrel cortex? Please clarify.

The reviewer’s point is well taken. We have added power spectra of the resting-state data which provides amplitude and frequency information. The integrated area under the curve of the power spectra is equal to the variance. Widefield-based measures of vasomotion were taken from the 1 mm ROI in the center of the whisker barrel cortex.

We have added the following text to the Materials & Methods (Line 560): “Variance during the resting-state for both ∆[HbT] and diameter signals (Fig. 7) was taken from resting-state events lasting ≥10 seconds in duration. Average ∆[HbT] from within the 1 mm ROI over the vibrissae representation of SI during each arousal state was taken with respect to awake resting baseline events ≥10 seconds in duration.”

(9) On page 13, the title seems like a bit strong. The data show a change in variance but that does not necessarily mean a change in absolute amplitude. Also, I did not see any reports of absolute vessel widths between groups from 2P experiments so any difference in the sampling of larger vs smaller arterioles could have affected the variance (ie. % changes could be much larger in smaller arterioles).

We have updated the title of Figure 7 to specifically state power (which is equivalent to the variance) rather than amplitude (Line 331). We have also added absolute vessel widths to the Results (Line 340): “There was no difference in resting-state (baseline) diameter between the groups, with Blank-SAP having a diameter of 24.4 ± 7.5 μm and SP-SAP having a diameter of 23.0 ± 9.4 μm (Fest, p ti 0.61). “

(10) Big picture question. How could a manipulation that affects so few cells in 1 hemisphere (below 0.5% of total neurons in a region comprising 1-2% of the volume of one hemisphere) have such profound effects in both hemispheres? The authors suggest that some may have long-range interhemispheric projections, but that is presumably a fraction of the already small fraction of Nos1 neurons. Perhaps these neurons have specializing projections to subcortical brain nuclei (Nucleus Basilis, Raphe, Locus Coerulus, reticular thalamus, etc) that then project widely to exert this outsized effect? Has there not been a detailed anatomical characterization of their efferent projections to cortical and sub-cortical areas? This point could be raised in the discussion.

We apologize for the lack of clarity of our work in this point. We would like to clarify that the only analysis showing a change in the unablated hemisphere being coherence/correlation analysis between the two hemispheres. Other metrics (LFP power and CBV power spectra) do not change in the hemisphere contralateral to the injections site, as we show in data added in two supplementary figures (Fig. S4 and 7). The coherence/correlation is a measure of the correlated dynamics in the two hemispheres. For this metric to change, there only needs to be a change in the dynamics of one hemisphere relative to another. If some aspects of the synchronization of neural and vascular dynamics across hemispheres are mediated by concurrent activation of type I nNOS neurons in both hemispheres, ablating them in one hemisphere will decrease synchrony. It is possible that type I nNOS neurons make some subcortical projections that were not reported in previous work (Tomioka 2005, Ruff 2024), but if these exist they are likely to be very small in number as they were not noted.

We have added the text in the Results (Line 228): “In contrast to the observed reductions in LFP in the ablated hemisphere, we noted no gross changes in the power spectra of neural LFP in the unablated hemisphere (Fig. S7) or power of the cerebral blood volume fluctuations in either hemisphere (Fig. S4).”

Line 335: “The variance in ∆[HbT] during rest, a measure of vasomotion amplitude, was significantly reduced following type-I nNOS ablation (Fig. 7a), dropping from 40.9 ± 3.4 μM^2^ in the Blank-SAP group (N ti 24, 12M/12F) to 23.3 ± 2.3 μM^2^ in the SP-SAP group (N ti 24, 11M/13F) (GLME p ti 6.9×10^-5^) with no significant difference in the unablated hemisphere (Fig. S7).”

**Reviewer #3 (Recommendations for the authors):**
(1) The reporting would be greatly strengthened by following ARRIVE guidelines 2.0: https://arriveguidelines.org/: aFrition rates and source of aFrition, justification for the use of 119 (beyond just consistent with previous studies), etc.

We performed a power analysis prior to our study aiming to detect a physiologically-relevant effect size of (Cohen’s d) ti 1.3, or 1.3 standard deviations from the mean. Alpha and Power were set to the standard 0.05 and 0.80 respectively, requiring around 8 mice per group (SP-SAP, Blank, and for histology, naïve animals) for multiple independent groups (ephys, GCamp, histology). To potentially account for any aFrition due to failures in Type-I nNOS neuron ablation or other problems (such as electrode failure or window issues) we conservatively targeted a dozen mice for each group. Of mice that were imaged (1P/2P), two SP-SAP mice were removed from the dataset (24 SP-SAP remaining) post-histological analysis due to not showing ablation of nNOS neurons, an aFrition rate of approximately 8%.

We have added the following text to the Materials & Methods (Line 441): “Sample sizes are consistent with previous studies (Echagarruga et al 2020, Turner et al 2023, Turner et al 2020, Zhang et al 2021) and based on a power analysis requiring 8-10 mice per group (Cohen’s d ti 1.3, α ti 0.05, (1 - β) ti 0.800). Experimenters were not blind to experimental conditions or data analysis except for histological experiments. Two SP-SAP mice were removed from the imaging datasets (24 SP-SAP remaining) due to not showing ablation of nNOS neurons during post-histological analysis, an aFrition rate of approximately 8%.”

(2) Intro, line 38: Description of the importance of neurovascular coupling needs improvement. Coordinated haemodynamic activity is vital for maintaining neuronal health and the energy levels needed.

We have added a sentence to the introduction (Line 41): “Neurovascular coupling plays a critical role in supporting neuronal function, as tightly coordinated hemodynamic activity is essential for meeting energy metabolism and maintaining brain health (Iadecola et al 2023, Schaeffer & Iadecola 2021).“

(3) Given the wide range of mice ages, how was the age accounted for/its effects examined?

Previous work from our lab has shown that there is no change in hemodynamics responses in awake mice over a wide range of ages (2-18 months), so the age range we used (3 and 9 months of age) should not impact this.

We have added the following text in the Results (Line 437): “Previous work from our lab has shown that the vasodilation elicited by whisker stimulation is the same in 2–4-month-old mice as in 18-month-old mice (BenneF, Zhang et al. 2024). As the age range used here is spanned by this time interval, we would not expect any age-related differences.”

(4) How was the susceptibility of low-frequency neuronal coupling signals to noise managed? How were the low-frequency bands results validated?

We are not sure what the referee is asking here. Our electrophysiology recordings were made differentially using stereotrodes with tips separated by ~100µm, which provides excellent common-mode rejection to noise and a localized LFP signal. Previous publications from our lab (Winder et al., Nature Neuroscience 2017; Turner et al., eLife2020) and others (Tu, Cramer, Zhang, eLife 2024) have repeatedly show that there is a very weak correlation between the power in the low frequency bands and hemodynamic signals, so our results are consistent with this previous work.

(5) It would be helpful to demonstrate the selectivity of cell *death* (as opposed to survival) induced by SP-SAP injections via assessments using markers of cell death.

We agree that this would be helpful complement to our histological studies that show loss of type-I nNOS neurons, but no loss of other cells and minimal inflammation with SP-saporin injections. However, we did not perform histology looking at cell death, only at surviving cells, given that we see no obvious inflammation or cells loss, which would be triggered by nonspecific cell death. Previous work has established that saporin is cytotoxic and specific only to cell that internalize the saporin. Internalization of saporin causes cell death via apoptosis (Bergamaschi, Perfe et al. 1996), and that the substance P receptor is internalized when the receptor is bound (Mantyh, Allen et al. 1995). Treatment of internalized saporin generates cellular debris that is phagocytosed by microglial, consistent with cell death (Seeger, Hartig et al. 1997). While it is possible that treatment of SP-saporin causes type 1 nNOS neurons to stop expressing nitric oxide synthase (which would make them disappear from our IHC staining), we think that this is unlikely given the literature shows internalized saporin is clearly cytotoxic.

We have added the following text to the Results (Line 131): “It is unlikely that the disappearance of type-I nNOS neurons is because they stopped expressing nNOS, as internalized saporin is cytotoxic. Exposure to SP-conjugated saporin causes rapid internalization of the SP receptor-ligand complex (Mantyh, Allen et al. 1995), and internalized saporin causes cell death via apoptosis (Bergamaschi, Perfe et al. 1996). In the brain, the resulting cellular debris from saporin administration is then cleared by microglia phagocytosis (Seeger, Hartig et al. 1997).”

(6) Was the decrease in inter-hemispheric correlation associated with any changes to the corpus callosum?

We noted no gross changes to the structure of the corpus callosum in any of our histological reconstructions following SSPSAP administration, however, we did not specifically test for this. Again, as we note in our reply in reviewer 2, the decrease in interhemispheric synchronization does not imply that there are changes in the corpus callosum and could be mediated by the changes in neural activity in the hemisphere in which the Type-I nNOS neurons were ablated.

(7) How were automated cell counts validated?

Criteria used for automated cell counts were validated with comparisons of manual counting as described in previous literature. We have added additional text describing the process in the Materials & Methods (Line 510): “For total cell counts, a region of interest (ROI) was delineated, and cells were automatically quantified under matched criteria for size, circularity and intensity. Image threshold was adjusted until absolute value percentages were between 1-10% of the histogram density. The function Analyze Par-cles was then used to estimate the number of particles with a size of 100-99999 pixels^2 and a circularity between 0.3 and 1.0 (Dao, Suresh Nair et al. 2020, Smith, Anderson et al. 2020, Sicher, Starnes et al. 2023). Immunoreactivity was quantified as mean fluorescence intensity of the ROI (Pleil, Rinker et al. 2015).”

(8) Given the weighting of the vascular IOS readout to the superficial tissue, it is important to qualify the extent of the hemodynamic contrast, ie the limitations of this readout.

We have added the following text to the Discussion (Line 385): “Intrinsic optical signal readout is primarily weighted toward superficial tissue given the absorption and scaFering characteristics of the wavelengths used. While surface vessels are tightly coupled with neural activity, it is still a maFer of debate whether surface or intracortical vessels are a more reliable indicator of ongoing activity (Goense et al 2012; Huber et al 2015; Poplawsky & Kim 2014).”

(9) Partial decreases observed through type-I iNOS neuronal ablation suggest other factors also play a role in regulating neural and vascular dynamics: data presented thus do *not* "indicate disruption of these neurons in diseases ranging from neurodegeneration to sleep disturbances," as currently stated. Please revise.

We agree with the reviewer. We have changed the abstract sentence to read (Line 30): “This demonstrates that a small population of nNOS-positive neurons are indispensable for regulating both neural and vascular dynamics in the whole brain, raising the possibility that loss of these neurons could contribute to the development of neurodegenerative diseases and sleep disturbances.”